# THINGS-data, a multimodal collection of large-scale datasets for investigating object representations in human brain and behavior

Martin N Hebart[1,2,3*†], Oliver Contier[2,4†], Lina Teichmann[1†], Adam H Rockter[1], Charles Y Zheng[5], Alexis Kidder[1], Anna Corriveau[1], Maryam Vaziri-Pashkam[1], Chris I Baker[1]

[1]Laboratory of Brain and Cognition, National Institute of Mental Health, National Institutes of Health, Bethesda, United States; [2]Vision and Computational Cognition Group, Max Planck Institute for Human Cognitive and Brain Sciences, Leipzig, Germany; [3]Department of Medicine, Justus Liebig University Giessen, Giessen, Germany; [4]Max Planck School of Cognition, Max Planck Institute for Human Cognitive and Brain Sciences, Leipzig, Germany; [5]Machine Learning Core, National Institute of Mental Health, National Institutes of Health, Bethesda, United States

*For correspondence:
hebart@cbs.mpg.de

†These authors contributed equally to this work

**Abstract** Understanding object representations requires a broad, comprehensive sampling of the objects in our visual world with dense measurements of brain activity and behavior. Here, we present THINGS-data, a multimodal collection of large-scale neuroimaging and behavioral datasets in humans, comprising densely sampled functional MRI and magnetoencephalographic recordings, as well as 4.70 million similarity judgments in response to thousands of photographic images for up to 1,854 object concepts. THINGS-data is unique in its breadth of richly annotated objects, allowing for testing countless hypotheses at scale while assessing the reproducibility of previous findings. Beyond the unique insights promised by each individual dataset, the multimodality of THINGS-data allows combining datasets for a much broader view into object processing than previously possible. Our analyses demonstrate the high quality of the datasets and provide five examples of hypothesis-driven and data-driven applications. THINGS-data constitutes the core public release of the THINGS initiative (https://things-initiative.org) for bridging the gap between disciplines and the advancement of cognitive neuroscience.

## Editor's evaluation

Hebart et al., present a landmark, multimodal massive dataset to support the study of visual object representation, including data measured from functional magnetic resonance imaging, magnetoencephalography, and behavioral similarity judgments. The compelling, condition-rich design, conducted over a thoughtfully curated and sampled set of object concepts will be highly valuable to the cognitive/computational/neuroscience community, yielding data that will be amenable to many empirical questions beyond the field of visual object recognition. The dataset is accompanied by quality control evaluations, as well as examples of analyses that the community can re-run and further explore for building new hypotheses that can be tested with such a rich dataset.

## Introduction

A central goal of cognitive neuroscience is to attain a detailed characterization of the recognition and understanding of objects in the world. Over the past few decades, there has been tremendous progress in revealing the basic building blocks of human visual and semantic object processing. For example, numerous functionally selective clusters have been identified in ventral and lateral occipito-temporal cortex that respond selectively to images of faces, scenes, objects, or body parts (*Downing et al., 2001*; *Epstein and Kanwisher, 1998*; *Kanwisher et al., 1997*; *Malach et al., 1995*). Likewise, several coarse-scale gradients have been revealed that span across these functionally selective regions and that reflect low-level visual properties such as eccentricity or curvature (*Arcaro et al., 2015*; *Groen et al., 2022*; *Yue et al., 2020*), mid-to-high-level properties such as animacy or size (*Caramazza and Shelton, 1998*; *Konkle and Caramazza, 2013*; *Konkle and Oliva, 2012*; *Kriegeskorte et al., 2008b*), or high-level semantics (*Huth et al., 2012*). These results have been complemented by studies in the temporal domain, revealing a temporal cascade of object-related responses that become increasingly invariant over time to visually specific features such as size and position (*Isik et al., 2014*), that reflect differences between visual and more abstract semantic properties (*Bankson et al., 2018*; *Cichy et al., 2014*; *Clarke et al., 2013*; *Clarke et al., 2015*), and that reveal the dynamics of feedforward and feedback processing (*Boring et al., 2022*; *Kietzmann et al., 2019*; *Mohsenzadeh et al., 2018*). These spatial and temporal patterns of object-related brain activity have been linked to categorization behavior (*Grootswagers et al., 2018*; *Ritchie et al., 2015*) and perceived similarity (*Bankson et al., 2018*; *Cichy et al., 2019*; *Mur et al., 2013*), indicating their direct relevance for overt behavior.

Despite these advances, our general understanding of the processing of visually-presented objects has remained incomplete. One major limitation stems from the enormous variability of the visual world and the thousands of objects that we can identify and distinguish (*Biederman, 1985*; *Hebart et al., 2019*). Different objects are characterized by a large and often correlated set of features (*Groen et al., 2017*; *Naselaris et al., 2021*), making it challenging to determine the overarching properties that govern the representational structure in visual cortex and behavior. A more complete understanding of visual and semantic object processing will almost certainly require a high-dimensional account (*Naselaris et al., 2021*; *Haxby et al., 2011*; *Hebart et al., 2020*; *Lehky et al., 2014*), which is impossible to derive from traditional experiments that are based only on a small number of stimuli or a small number of categories. Likewise, even large-scale datasets remain limited in the insights they can yield about object representations when they lack a systematic sampling of object categories and images.

To overcome these limitations, here we introduce THINGS-data, which consists of three multimodal large-scale datasets of brain and behavioral responses to naturalistic object images. There are three key aspects of THINGS-data that maximize its utility and set it apart from other large-scale datasets using naturalistic images (*Allen et al., 2022*; *Chang et al., 2019*; *Horikawa and Kamitani, 2017*; *Kay et al., 2008*). First, THINGS-data is unique in that it offers a broad, comprehensive and systematic sampling of object representations for up to 1854 diverse nameable manmade and natural object concepts. This is in contrast to previous large-scale neuroimaging datasets that focused primarily on dataset size, not sampling, and that often contain biases towards specific object categories (*Allen et al., 2022*; *Chang et al., 2019*). Second, THINGS-data is multimodal, containing functional MRI, magnetoencephalography (MEG) and behavioral datasets allowing analyses of both the spatial patterns and temporal dynamics of brain responses (*Ghuman and Martin, 2019*) as well as their relationship to behavior. In particular, THINGS-data comes with 4.70 million behavioral responses that capture the perceived similarity between objects with considerable detail and precision. Third, the THINGS database of object concepts and images (*Hebart et al., 2019*) comes with a growing body of rich annotations and metadata, allowing for direct comparisons of representations across domains, an extension to other methods and species (*Kriegeskorte et al., 2008a*), streamlined incorporation of computational modeling frameworks (*Kriegeskorte and Douglas, 2018*), and direct testing of diverse hypotheses on these large-scale datasets.

In this paper, we provide a detailed account of all aspects of THINGS-data, from acquisition and data quality checks to exemplary analyses demonstrating the potential utility of the data. These exemplary analyses primarily serve to highlight potential research directions that could be explored with these data. In addition, the analyses of the neuroimaging data reveal high reliability of findings across

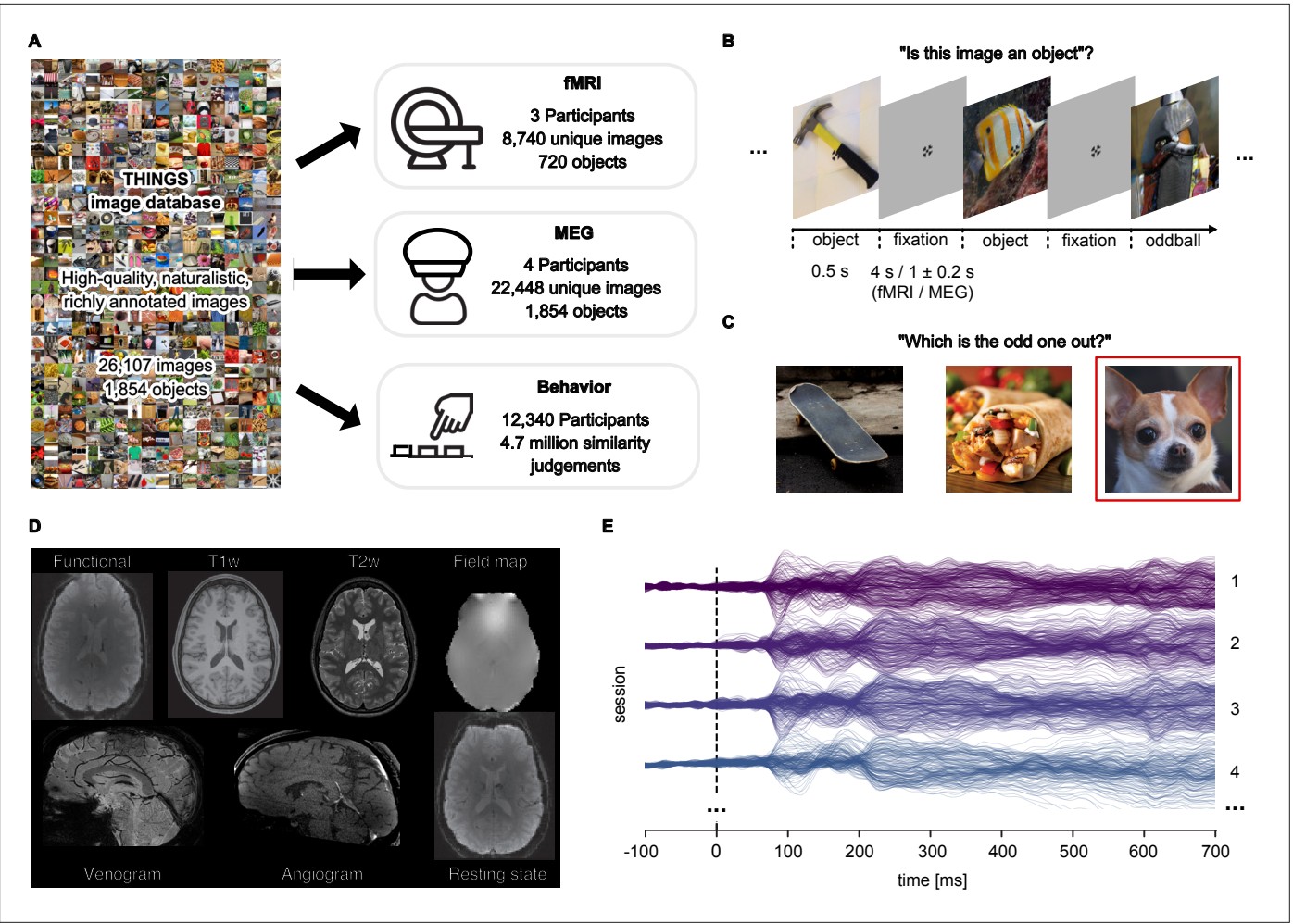

**Figure 1.** Overview over datasets. (**A**) THINGS-data comprises MEG, fMRI and behavioral responses to large samples of object images taken from the THINGS database. (**B**) In the fMRI and MEG experiment, participants viewed object images while performing an oddball detection task (synthetic image). (**C**) The behavioral dataset comprises human similarity judgements from an odd-one-out task where participants chose the most dissimilar object amongst three options. (**D**) The fMRI dataset contains extensive additional imaging data. (**E**) The MEG dataset provides high temporal resolution of neural response measurements in 272 channels. The butterfly plot shows the mean stimulus-locked response in each channel for four example sessions in one of the participants.

The online version of this article includes the following figure supplement(s) for figure 1:

**Figure supplement 1.** Effects of ICA denoising on fMRI noise ceiling estimates, for all three fMRI participants.

individual participants, underscoring the utility of densely sampling a small number of individuals. Finally, they replicate a large number of research findings, suggesting that these data can be used for revealing new insights into visual and semantic processing in human brain and behavior. We expect that THINGS-data will serve as an important resource for the community, enabling novel analyses to provide significant insights into visual object processing as well as validation and extension of existing findings. THINGS-data reflects the core release of datasets as part of the THINGS initiative (https://things-initiative.org), which will provide numerous multimodal and multispecies behavioral, neurophysiology, and neuroimaging datasets based on the same images, offering an important general resource that bridges the gap between disciplines for the advancement of the cognitive neurosciences.

# Results

## A multimodal collection of datasets of object representations in brain and behavior

We collected three datasets that extensively sampled object representations using functional MRI (fMRI), magnetoencephalography (MEG), and behavior (*Figure 1*). To this end, we drew on the THINGS database (*Hebart et al., 2019*), a richly-annotated database of 1854 object concepts representative of the American English language which contains 26,107 manually curated naturalistic object images. The comprehensive set of object categories, the large number of high-quality naturalistic images, and the rich set of semantic and image annotations make THINGS ideally suited for the large-scale collection of imaging and behavioral datasets.

During the fMRI and MEG experiments, participants were shown a representative subset of THINGS images, spread across 12 separate sessions (fMRI: N=3, 8740 unique images of 720 objects; MEG: N=4, 22,448 unique images of 1854 objects). Images were shown in fast succession (fMRI: 4.5 s; MEG: 1.5±0.2 s; *Figure 1B*), and participants were instructed to maintain central fixation. Please note that for the MEG and fMRI experiments, we chose non-overlapping sets of participants to ensure they had not seen individual images before and thus to minimize potential memory effects on measured object representations. To ensure engagement, participants performed an oddball detection task responding to occasional artificially-generated images. A subset of images (fMRI: n=100; MEG: n=200) were shown repeatedly in each session to estimate noise ceilings (*Lage-Castellanos et al., 2019*) and to provide a test set for model evaluation (see Appendix 1 for details on the concept and image selection strategy).

Beyond the core functional imaging data in response to THINGS images, additional structural and functional imaging data as well as eye-tracking and physiological responses were gathered. Specifically, for MEG, we acquired T1-weighted MRI scans to allow for cortical source localization. Eye movements were monitored in the MEG to ensure participants maintained central fixation (see Appendix 2 and *Appendix 2—figure 1* for extensive eye-movement related analyses). For MRI, we collected high-resolution anatomical images (T1- and T2-weighted), measures of brain vasculature (Time-of-Flight angiography, T2*-weighted), and gradient-echo field maps. In addition, we ran a functional localizer to identify numerous functionally specific brain regions, a retinotopic localizer for estimating population receptive fields, and an additional run without external stimulation for estimating resting-state functional connectivity. Finally, each MRI session was accompanied by physiological recordings (heartbeat and respiration) to support data denoising. Based on these additional data, we computed a variety of data derivatives for users to refine their analyses. These derivatives include cortical flat-maps which allow for visualizing statistical results on the entire cortical surface (*Gao et al., 2015*), independent-component based noise regressors which can be used for improving the reliability of obtained results, regions of interest for category-selective and early visual brain areas which allow for anatomically-constrained research questions, and estimates of retinotopic parameters, such as population receptive field location and size.

THINGS-data also includes 4.70 million human similarity judgements collected via online crowd-sourcing for 1854 object images. In a triplet odd-one-out task, participants (N=12,340) were presented with three objects from the THINGS database and were asked to indicate which object is the most dissimilar. The triplet odd-one-out task assesses the similarity of two objects in the context imposed by a third object. With a broad set of objects, this offers a principled approach for measuring context-independent perceived similarity with minimal response bias, but also allows for estimating context-dependent similarity, for example by constraining similarity to specific superordinate categories, such as animals or vehicles. An initial subset of 1.46 million of these odd-one-out judgments were reported in previous work (*Hebart et al., 2020*; *Zheng et al., 2019*), and the 4.70 million trials reported here represent a substantial increase in dataset size and the ability to draw inferences about fine-grained similarity judgments. Beyond dataset size, two notable additions are included. First, we collected age information, providing a cross-sectional sample for how mental representations may change with age. Second, we collected a set of 37,000 within-subject triplets to estimate variability at the subject level. Taken together, the behavioral dataset provides a massive set of perceived similarity judgements of object images and can be linked to neural responses measured in MEG and fMRI, opening the door to studying the neural processes underlying perceived similarity at scale, for a wide range of objects.

The remaining results section will be structured as follows: We will first describe the quality and reliability of both neuroimaging datasets, followed by the description of the quality of the behavioral dataset. Then, we will showcase the validity and suitability of the datasets for studying questions about behavioral and neural object representations. This will include multivariate pairwise decoding of hundreds of object categories, encoding analyses serving as a large-scale replication of the animacy and size organization in occipitotemporal cortex, representational similarity analysis of patterns of brain activity and perceived similarity, and a novel MEG-fMRI fusion approach based on directly regressing MEG responses onto fMRI voxel activation patterns.

## Data quality and data reliability in the fMRI and MEG datasets

To be useful for addressing diverse research questions, we aimed at providing neuroimaging datasets with excellent data quality and high reliability. To reduce variability introduced through head motion and alignment between sessions, fMRI and MEG participants wore custom head casts throughout all sessions. *Figure 2* demonstrates that overall head motion was, indeed, very low in both neuroimaging datasets. In the fMRI dataset, the mean framewise displacement per run was consistently below 0.2 mm. In the MEG, head position was recorded between runs and showed consistently low head motion for all participants during sessions (median <1.5 mm). Between sessions, changes in MEG head position were slightly higher but remained overall low (median <3 mm). A visual comparison of the evoked responses for each participant across sessions in different sensor groups highlights that the extent of head motion we observed does not appear to be detrimental for data quality (see *Figure 2—figure supplement 1*).

To further improve fMRI data quality and provide easily usable data, we conducted two additional processing steps. First, since fMRI data contains diverse sources of noise including head motion, pulse, heartbeat, and other sources of physiological and scanner-related noise, we developed a custom denoising method based on independent component analysis (*Beckmann and Smith, 2004*), which involved hand-labeling a subset of components and a set of simple heuristics to separate signal from noise components (see Methods for details). This approach yielded strong and consistent improvements in the reliability of single trial BOLD response estimates (*Figure 1—figure supplement 1*). Second, we estimated the BOLD response amplitude to each object image by fitting a single-trial regularized general linear model on the preprocessed fMRI time series with voxel-specific estimates of the HRF shape (see Methods). Together, these methods yielded much higher data reliability and provided a format that is much smaller than the original time series and that is amenable to a wider range of analysis techniques, including data-driven analyses. This reduced set of BOLD parameter estimates is used for all analyses showcased in this manuscript and is part of the publicly available data (see Data availability).

To provide a quantitative assessment of the reliability of the fMRI and MEG datasets, we computed noise ceilings. Noise ceilings are defined as the maximum performance any model can achieve given the noise in the data (*Lage-Castellanos et al., 2019*) and are based on the variability across repeated measurements. Since noise ceiling estimates depend on the number of trials averaged in a given analysis, we estimated them separately for the 12 trial repeats of the test set and for single trial estimates. Noise ceilings in the test set were high (*Figure 2*), with up to 80% explainable variance in early visual cortex for fMRI (*Figure 2C*) and up to 70% explainable variance in MEG (*Figure 2D*, *Figure 2—figure supplement 2*). Individual differences between participants indicated that performance was particularly high for fMRI participants F1 and F2 and MEG participants M2 and M3 but qualitatively similar for all participants. For single-trial estimates, as expected, noise ceilings were lower and varied more strongly across participants (*Figure 2—figure supplement 3*). This suggests that these neuroimaging datasets are ideally suited for analyses that incorporate estimates across multiple trials, such as encoding or decoding models or data-driven analyses at the level of object concepts.

## Data quality and data reliability in the behavioral odd-one out dataset: A 66-dimensional embedding captures fine-grained perceived similarity judgments

To achieve a full estimate of a behavioral similarity matrix for all 1854 objects, we would have to collect 1.06 billion triplet odd-one-out judgments. We previously demonstrated (*Hebart et al., 2020*) that 1.46 million trials were sufficient to generate a sparse positive similarity embedding (SPoSE) (*Zheng*

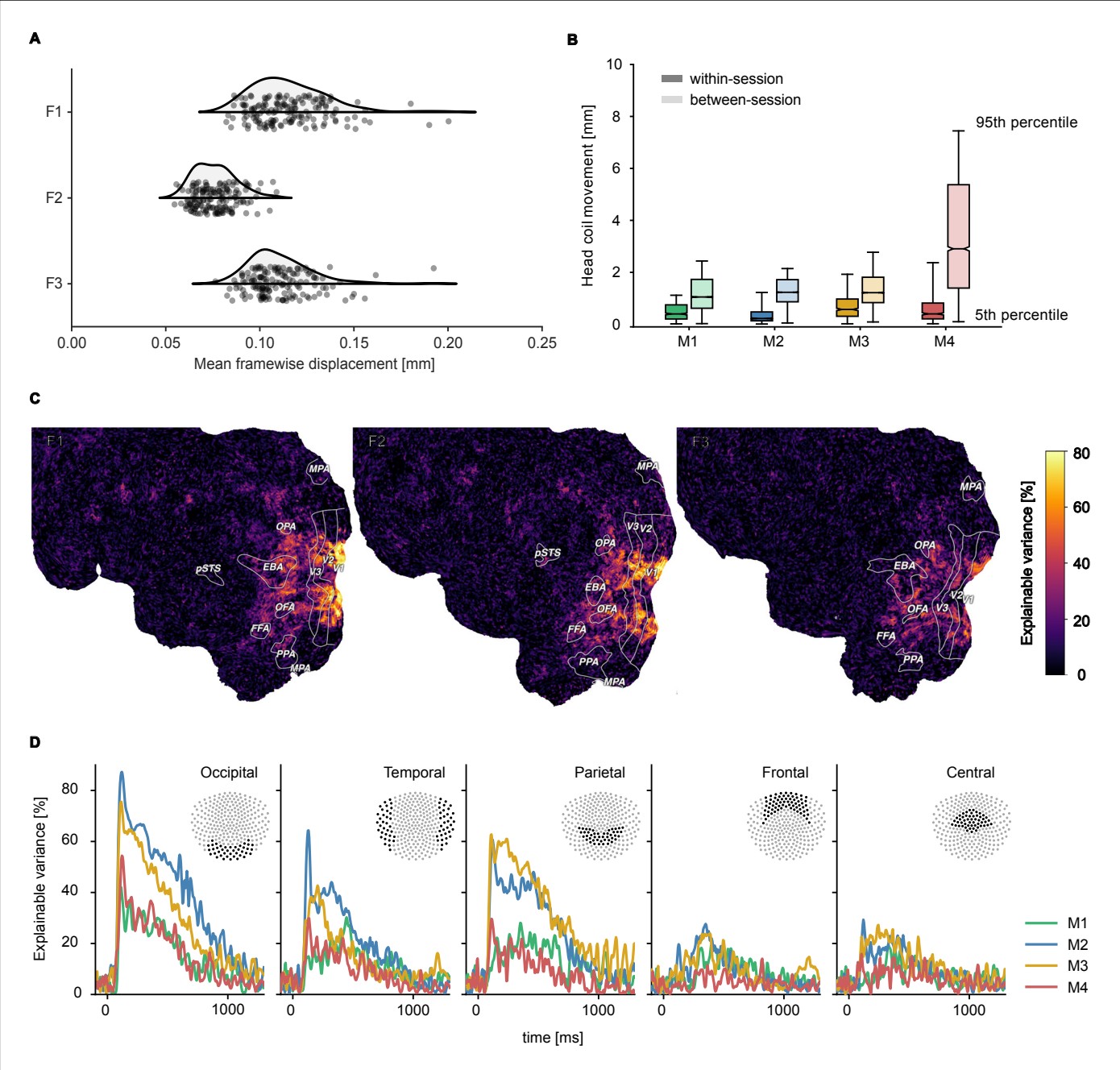

**Figure 2.** Quality metrics for fMRI and MEG datasets. fMRI participants are labeled F1-F3 and MEG participants M1-M4 respectively. (**A**) Head motion in the fMRI experiment as measured by the mean framewise displacement in each functional run of each participant. (**B**) Median change in average MEG head coil position as a function of the Euclidean distance of all pairwise comparisons between all runs. Results are reported separately for comparisons within sessions and between sessions (see *Figure 2—figure supplement 4* for all pairwise distances). (**C**) fMRI voxel-wise noise ceilings in the test dataset as an estimate of explainable variance visualized on the flattened cortical surface. The labeled outlines show early visual (V1–V3) and category-selective brain regions identified based on the population receptive field mapping and localizer data, respectively. (**D**) MEG time-resolved noise ceilings similarly show high reliability, especially for occipital, parietal, and temporal sensors.

The online version of this article includes the following figure supplement(s) for figure 2:

**Figure supplement 1.** Event-related fields for occipital, temporal, and parietal sensors.

**Figure supplement 2.** MEG noise ceilings for all sensors.

**Figure supplement 3.** fMRI voxel-wise noise ceilings per participant projected onto the flattened cortical surface.

**Figure supplement 4.** Head coil positioning across runs in the MEG experiment.

**Figure supplement 5.** Example visualization used for the manual labeling of independent components.

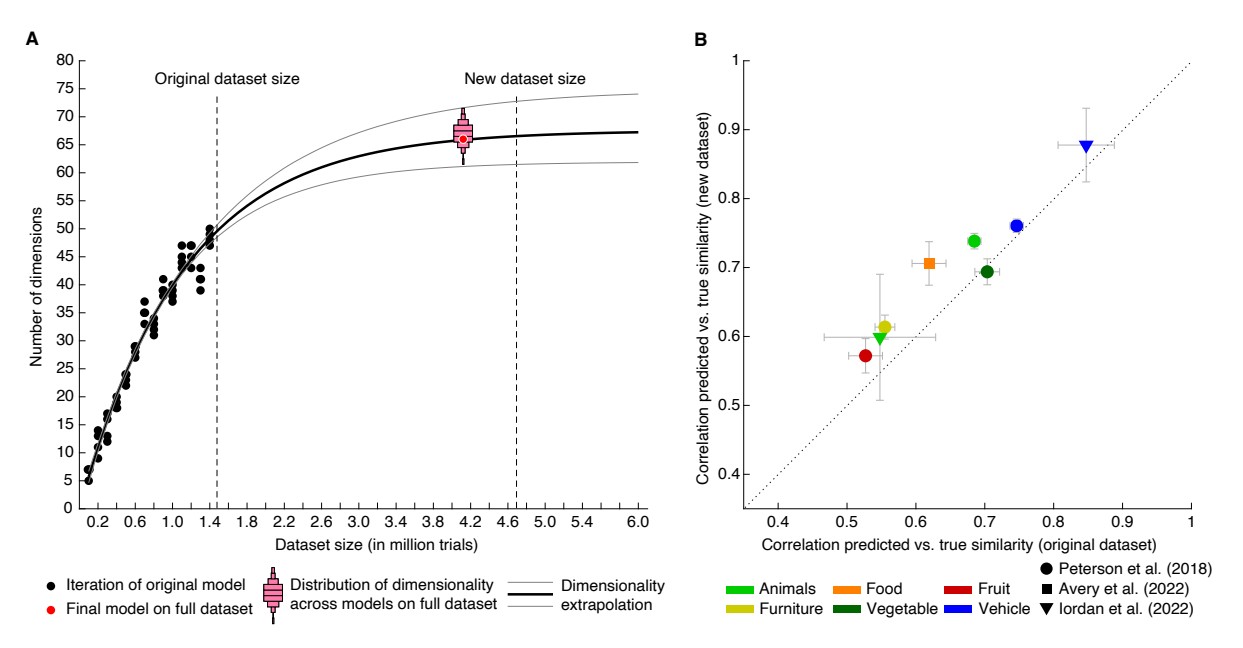

**Figure 3.** Behavioral similarity dataset. (**A**) How much data is required to capture the core representational dimensions underlying human similarity judgments? Based on the original dataset of 1.46 million triplets (*Hebart et al., 2020*), it was predicted that around 4.5–5 million triplets would be required for the curve to saturate. Indeed, for the full dataset, the dimensionality was found to be 66, in line with the extrapolation. Red bars indicate histograms for dimensionality across several random model initializations, while the final model was chosen to be the most stable among this set. (**B**) Within-category pairwise similarity ratings were predicted better for diverse datasets using the new, larger dataset of 4.70 million triplets (4.10 million training samples), indicating that this dataset contains more fine-grained similarity information. Error bars reflect standard errors of the mean.

The online version of this article includes the following figure supplement(s) for figure 3:

**Figure supplement 1.** Changes in embedding dimensions between original embedding (49 dimensions) and the new embedding (66 dimensions) based on the full dataset.

*et al., 2019*) that approached noise ceiling in predicting choices in left-out trials and pairwise similarity. SPoSE yielded 49 interpretable behavioral dimensions reflecting perceptual and conceptual object properties (e.g. colorful, animal-related) and thus identified what information may be used by humans to judge the similarity of objects in this task. Yet, several important questions about the general utility of these data could not be addressed with this original dataset.

First, how much data is enough to capture the core dimensions underlying human similarity judgments? Previously, we had shown that performance of our embedding at predicting triplet choices had saturated even with the original 1.46 million trials, yet dimensionality continued to increase with dataset size (*Hebart et al., 2020*). Before collecting additional data and using different subsets of the original dataset, we estimated that model dimensionality would saturate around 67.5 dimensions and would reach ~66.5 dimensions for 4.5–5 million trials (*Figure 3A*). Indeed, when re-running the model with the full dataset of 4.70 million trials (4.10 million for training), embedding dimensionality turned out as predicted: from a set of 72 randomly-initialized models, we chose the most reliable embedding as the final embedding, revealing 66 interpretable dimensions underlying perceived similarity judgments (see Methods for details). Thus, increasing dataset size beyond this large dataset may no longer yield noticeable improvements in predictive performance or changes in embedding dimensionality at the global level of similarity, and potential improvements may not justify the cost of collecting additional data. Thus, rather than continuing to increase dataset size, future research on representational object dimensions may focus more strongly on individual differences, within category similarity, different sensory domains, or abstracted stimuli.

In the final 66-dimensional embedding, many dimensions were qualitatively very similar to the original 49 dimensions (*Figure 3—figure supplement 1*), and some new dimensions were splits derived from previously mixed dimensions (e.g. plant-related and green) or highlighted more fine-grained aspects of previous dimensions (e.g. dessert rather than food). Overall model performance was similar

yet slightly lower for the new and larger as compared to the original and smaller dataset (original: 64.60 ± 0.23%, new: 64.13 ± 0.18%), while noise ceilings were comparable (original noise ceiling dataset: 68.91 ± 1.07%, new noise ceiling datasets: 68.74 ± 1.07% and 67.67 ± 1.08%), indicating that the larger dataset was of similar quality. However, these noise ceilings were based on between-subject variability, leaving open a second question: how strongly did within-subject variability contribute to overall variability in the data? To estimate the within-subject noise ceiling, we inspected the consistency of within-subject triplet repeats. The within-subject noise ceiling was at 86.34 ± 0.46%. Even though this estimate constitutes an upper bound of the noise ceiling, since identical trials were repeated after only 16–20 triplets to compute reliability estimates, these results indicate that a lot of additional variance may be captured when accounting for differences between individuals. Thus, participant-specific modeling based on this new large-scale behavioral dataset may yield additional, novel insights into the nature of mental object representations.

Third, while increases in dataset size did not lead to notable improvements in overall performance, did increasing the dataset size improve more fine-grained predictions of similarity? To address this question, we used several existing datasets of within-category similarity ratings (*Avery et al., 2022*; *Iordan et al., 2022*; *Peterson et al., 2018*) and computed similarity predictions. Rather than computing similarity across all possible triplets, these predictions were constrained to triplet contexts within superordinate categories (e.g. animals, vehicles). We expected the overall predictive performance to vary, given that these existing similarity rating datasets were based on a different similarity task or used different images. Yet, improvements are expected if fine-grained similarity can be estimated better with the large dataset than the original dataset. Indeed, as shown in *Figure 3B*, seven out of eight datasets showed an improvement in predicted within-category similarity (mean improvement M=0.041 ± 0.007, p<0.001, bootstrap difference test). This demonstrates that within-category

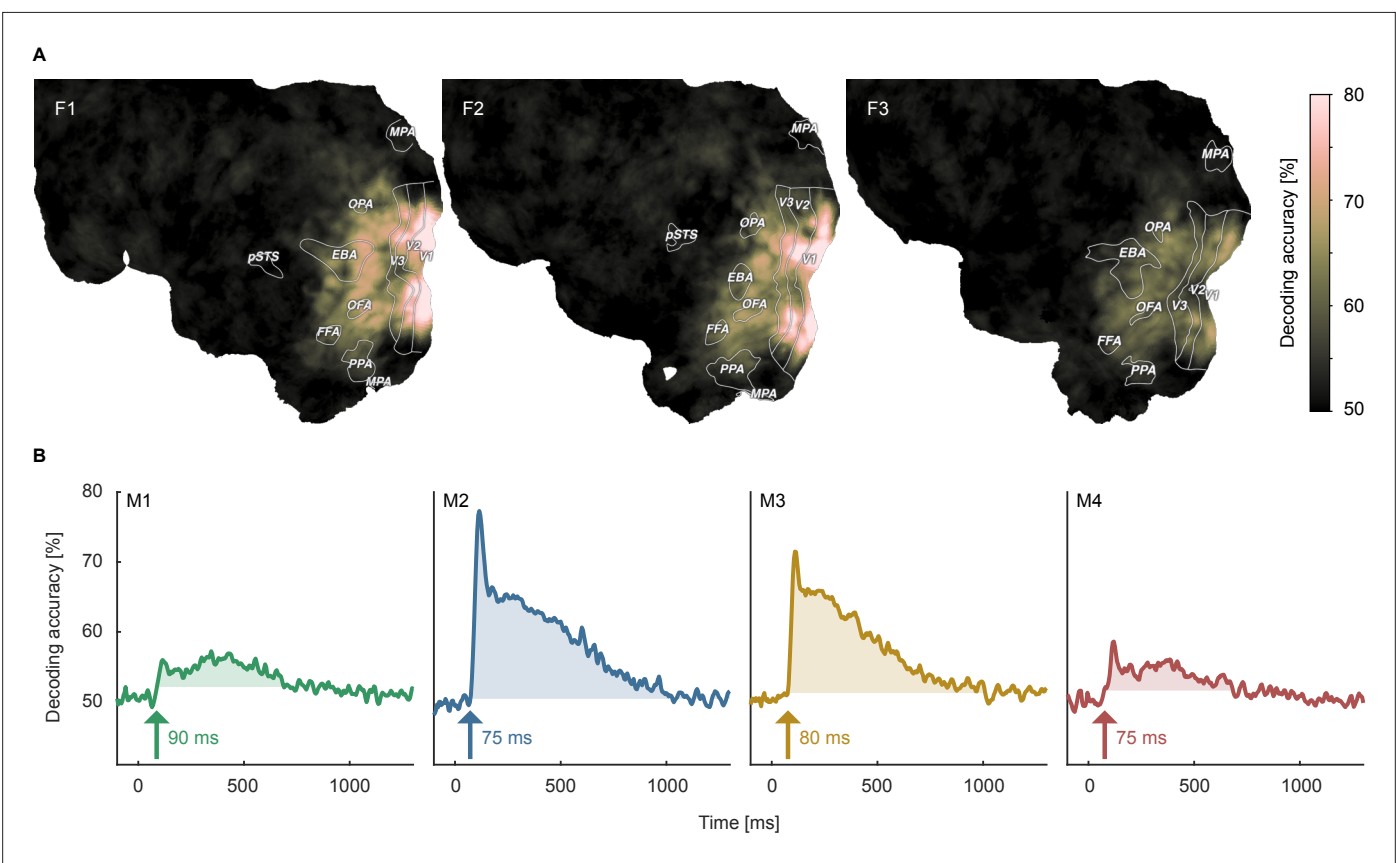

**Figure 4.** Object image decoding in fMRI and MEG. (**A**) Decoding accuracies in the fMRI data from a searchlight-based pairwise classification analysis visualized on the cortical surface. (**B**) Analogous decoding accuracies in the MEG data plotted over time. The arrow marks the onset of the largest time window where accuracies exceed the threshold which was defined as the maximum decoding accuracy observed during the baseline period.

similarity could be estimated more reliably with the larger dataset, indicating that the estimated embedding indeed contained more fine-grained information.

## Robustly decodable neural representations of objects

Having demonstrated the quality and overall reliability of the neuroimaging datasets, we aimed at validating their general suitability for studying questions about the neural representation of objects. To this end, we performed multivariate decoding on both the fMRI and MEG datasets, both at the level of individual object images, using the repeated image sets, and at the level of object category, using the 12 example images per category. Demonstrating the possibility to decode image identity and object category thus serves as a baseline analysis for more specific future research analyses.

When decoding the identity of object images, for fMRI we found above chance decoding accuracies in all participants throughout large parts of early visual and occipitotemporal cortices (*Figure 4A*), with peak accuracies in early visual cortex, reaching 80% in participants F1 and F2. In MEG, we found above-chance decoding within an extended time-window (~80–1000ms) peaking ~100ms after stimulus onset, approaching 70–80% in participants M2 and M3 (*Figure 4B*).

Moving from the level of decoding of individual images to the decoding of object category, for fMRI, accuracies now peaked in high-level visual cortex (*Figure 5A*). Likewise, for MEG the early decoding accuracies were less pronounced in absolute magnitude as compared to object image decoding (*Figure 5C & D*). Together, these results confirm that both object image and object category can be read out reliably from both neuroimaging datasets, demonstrating their general usefulness for addressing more specific research questions about object identity.

To demonstrate the utility of the datasets for exploring the representational structure in the neural response patterns evoked by different object categories, we additionally visualized their relationships in a data-driven fashion using multidimensional scaling (MDS) and highlighted clusters formed by superordinate categories. In fMRI, spatial response patterns across voxels in object-selective brain areas formed distinct clusters for the superordinate categories animals vs. food (*Figure 5B*). MEG sensor patterns showed differences between categorical clustering at early and late time points (e.g. early differences for vehicles vs. tools, late differences for animals vs. food), indicating that information about superordinate categories arise at different times (*Figure 5E*).

## Large-scale replication of experimental findings: The case of animacy and size

The large-scale neuroimaging datasets can be used for addressing an abundance of new questions by pairing them with existing or new metadata for object categories, object concepts, or object images. However, they can also be used to test the degree to which previously shown findings hold for a broader set of objects. To this end, we aimed at replicating the seminal findings of cortical gradients of animacy and size tuning in occipitotemporal cortex (*Konkle and Caramazza, 2013*; *Konkle and Oliva, 2012*; *Welbourne et al., 2021* ) and the temporal dynamics of object animacy and size representation (*Grootswagers et al., 2019*; *Khaligh-Razavi et al., 2018*; *Wang et al., 2022*). We used animacy and size ratings available for each object in the THINGS concept metadata (*Stoinski et al., 2022*) and used them to predict single-trial fMRI and MEG responses.

In line with previous findings, the fMRI results (*Figure 6A*, *Figure 6—figure supplements 1 and 2*) replicated the well-known alternating and spoke-like preference for animate vs. inanimate and small vs. big objects in occipitotemporal cortex (*Konkle and Caramazza, 2013*). As expected, we found a strong preference for animate objects in fusiform gyrus and a transition along the mid-fusiform sulcus to inanimate preference in parahippocampal cortex (*Grill-Spector and Weiner, 2014*). Regarding real-world size, place-selective brain areas (parahippocampal place area, occipital place area, and medial place area) showed a preference for big objects, and sections of lateral occipital cortex, which partly overlap with the extrastriate body area, showed a preference for small objects. While the results so far replicate the known topography of object animacy and size, in contrast to previous studies (*Konkle and Caramazza, 2013*; *Konkle and Oliva, 2012*; *Welbourne et al., 2021*), we found a preference for large objects in parts of the fusiform gyrus, as well as a preference for small objects in a stretch of cortex in-between fusiform and parahippocampal gyrus. While the reasons for these diverging results are unclear, previous studies used a smaller range of sizes, and objects in the present dataset excluded certain stimuli that serve the purpose of navigation (e.g. houses) or that tend to be small (e.g. food),

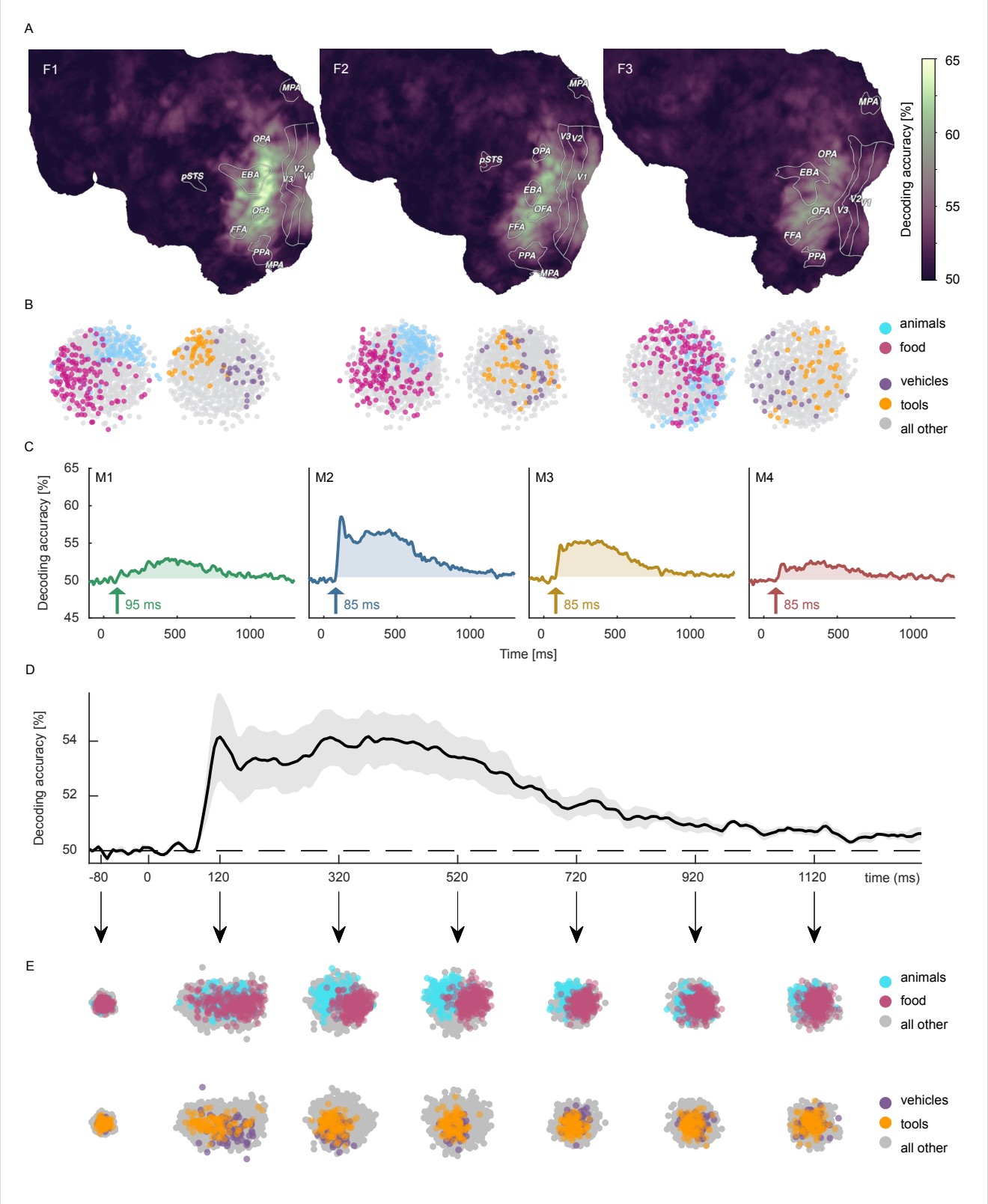

**Figure 5.** Object category decoding and multidimensional scaling of object categories in fMRI and MEG. (**A**) Decoding accuracies in the fMRI data from a searchlight-based pairwise classification analysis visualized on the cortical surface. (**B**) Multidimensional scaling of fMRI response patterns in occipito-temporal category-selective brain regions for each individual subject. Each data point reflects the average response pattern of a given object category. Colors reflect superordinate categories. (**C**) Pairwise decoding accuracies of object category resolved over time in MEG for each individual subject. (**D**)

*Figure 5 continued on next page*

*Figure 5 continued*

Group average of subject-wise MEG decoding accuracies. Error bars reflect standard error of the mean across participants (n = 4). (**E**) Multidimensional scaling for the group-level response pattern at different timepoints. Colors reflect superordinate categories and highlight that differential responses can emerge at different stages of processing.

which may have affected these results. Disentangling the functional topography of object size at these different scales is a subject for future research.

With regard to the temporal dynamics, our data support previous findings (*Cichy et al., 2014*; *Ritchie et al., 2015*; *Grootswagers et al., 2019*; *Khaligh-Razavi et al., 2018*; *Wang et al., 2022*; *Carlson et al., 2013*; *Grootswagers et al., 2017*). For animacy, previous small-scale studies varied in reported decoding peaks between 140 and 350ms, with most results around 140–190ms. Our large-scale data corroborate this overall trend, showing a pronounced peak for animacy information at ~180ms in all participants (*Figure 6B*). Similarly, object size information was reliably present in the neural signal for all participants, albeit weaker than animacy and peaking later, further supporting previous findings. Thus, while previous findings were based on a small number of objects cropped from their natural background, our data generalize these findings by demonstrating that they also hold for a comprehensive range of thousands of objects and by extending previous findings to object images embedded in a natural background.

## Linking object representations between fMRI, MEG, and behavior

To demonstrate avenues for integrating neuroimaging and behavioral datasets, we performed representational similarity analysis (*Kriegeskorte et al., 2008a*) to identify how well human similarity judgements reflected spatial and temporal brain responses. To this end, we correlated the behavioral similarity matrix with similarity matrices derived from fMRI searchlight voxel patterns across space and MEG sensor patterns across time. For fMRI, we found representational similarities in large parts of occipito-temporal cortex, with the strongest correspondence in ventral temporal and lateral occipital areas (*Figure 7A*), in line with previous findings (*Cichy et al., 2019*). For MEG, representational similarities with behavior were present as early as 80–100ms after stimulus onset in all participants, which is earlier than reported in previous studies (*Bankson et al., 2018*; *Cichy et al., 2019*). Correlations exceeding the maximum value during the baseline period were sustained in all participants for at least 500ms (*Figure 7B*). Together, these results showcase how the behavioral and neuroimaging data can be linked for studying the large-scale cortical topography and temporal response dynamics underlying subjectively perceived object similarities, from small sets of individual objects all the way to a comprehensive evaluation based on thousands of objects.

## Direct regression-based MEG-fMRI fusion

One advantage of a multimodal collection of datasets is that we can combine fMRI and MEG to reveal the spatiotemporal dynamics underlying object processing. An existing popular approach for combining MEG and fMRI (*Cichy and Oliva, 2020*) relies on correlating representational dissimilarity matrices (RDMs) obtained from fMRI for specific ROIs with time-resolved RDMs recorded with MEG. Thus, while this approach allows for comparisons at the population level both for MEG and fMRI, it is indirect and introduces additional assumptions about the spatial distribution of activity patterns and their representational similarity metric. Specifically, MEG-fMRI fusion based on classical representational similarity analysis (RSA; *Kriegeskorte et al., 2008a*) requires the a priori selection of sensors and/or voxels to include into the computation of an RDM, additionally assumes that all voxels and MEG sensors contribute equally to the representational similarity (*Kaniuth and Hebart, 2022*), and requires the selection of a similarity metric (*Bobadilla-Suarez et al., 2020*; *Ramírez et al., 2020*). In contrast, the size of THINGS-data allows using the MEG data directly to predict responses in fMRI ROIs or even individual voxels without having to rely on these assumptions. To showcase this analysis approach, we focused on two ROIs, V1 and FFA, and predicted average ROI responses recorded with fMRI from time-resolved multivariate pattern responses recorded with MEG using conventional multiple linear regression (*Figure 8*).

The results from all four MEG participants showed that V1 responses could be predicted by MEG activity starting within the first 100ms, corresponding to earlier MEG work (*Cichy et al., 2014*; *Cichy et al., 2015*) and work in non-human primates (*Bullier, 2001*; *Schmolesky et al., 1998*). In contrast, the

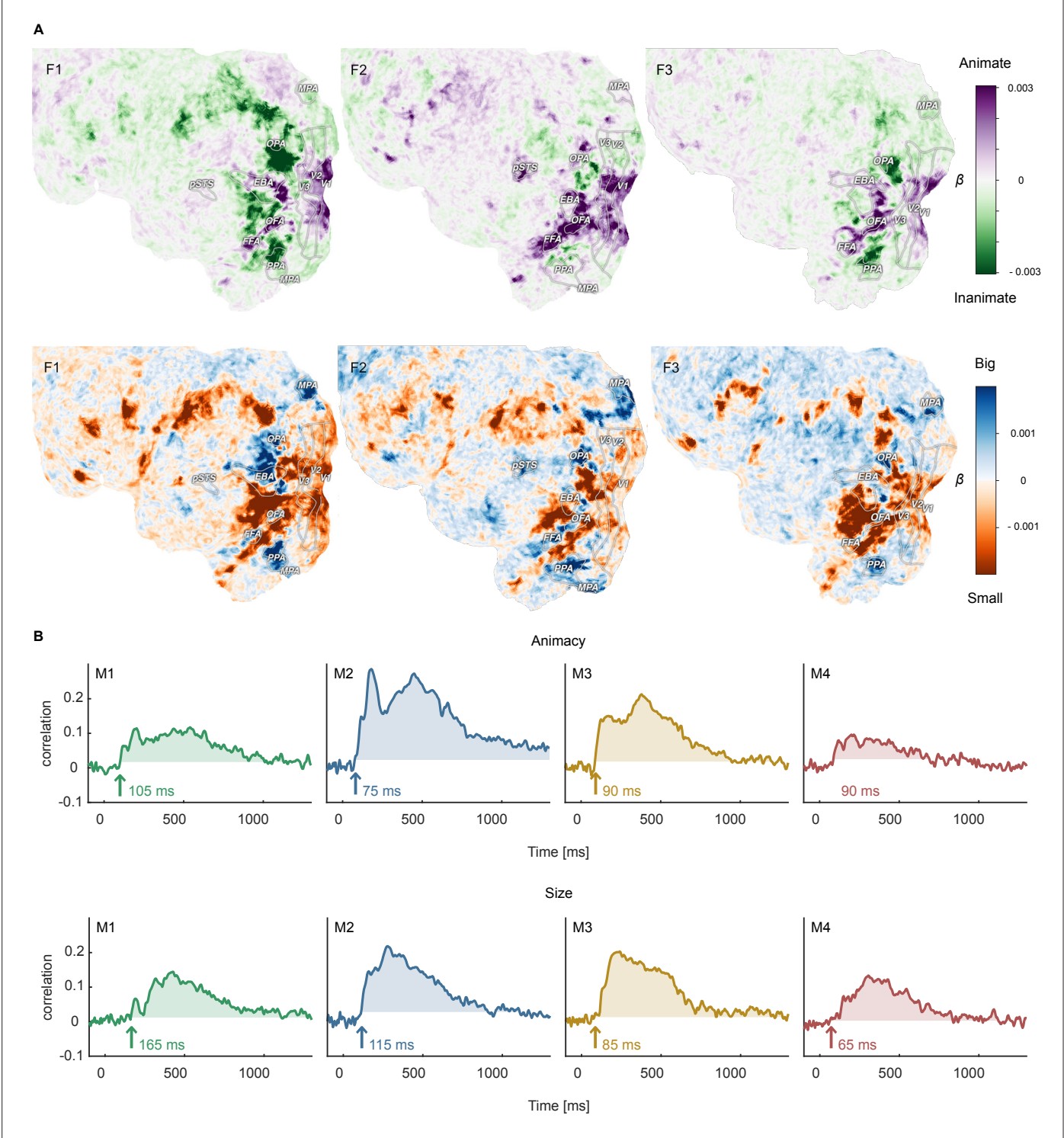

**Figure 6.** Functional topography and temporal dynamics of object animacy and size. (**A**) Voxel-wise regression weights for object animacy and size as predictors of trial-wise fMRI responses. The results replicate the characteristic spoke-like topography of functional tuning to animacy and size in occipitotemporal cortex. (**B**) Time courses for the animacy (top) and size (bottom) information in the MEG signal. The time courses were obtained from a cross-validated linear regression and show the correlation between the predicted and true animacy and size labels. Shaded areas reflect the largest time window exceeding the maximum correlation during the baseline period.

The online version of this article includes the following figure supplement(s) for figure 6:

**Figure supplement 1.** Functional topography of object animacy.

**Figure supplement 2.** Functional topography of object size.

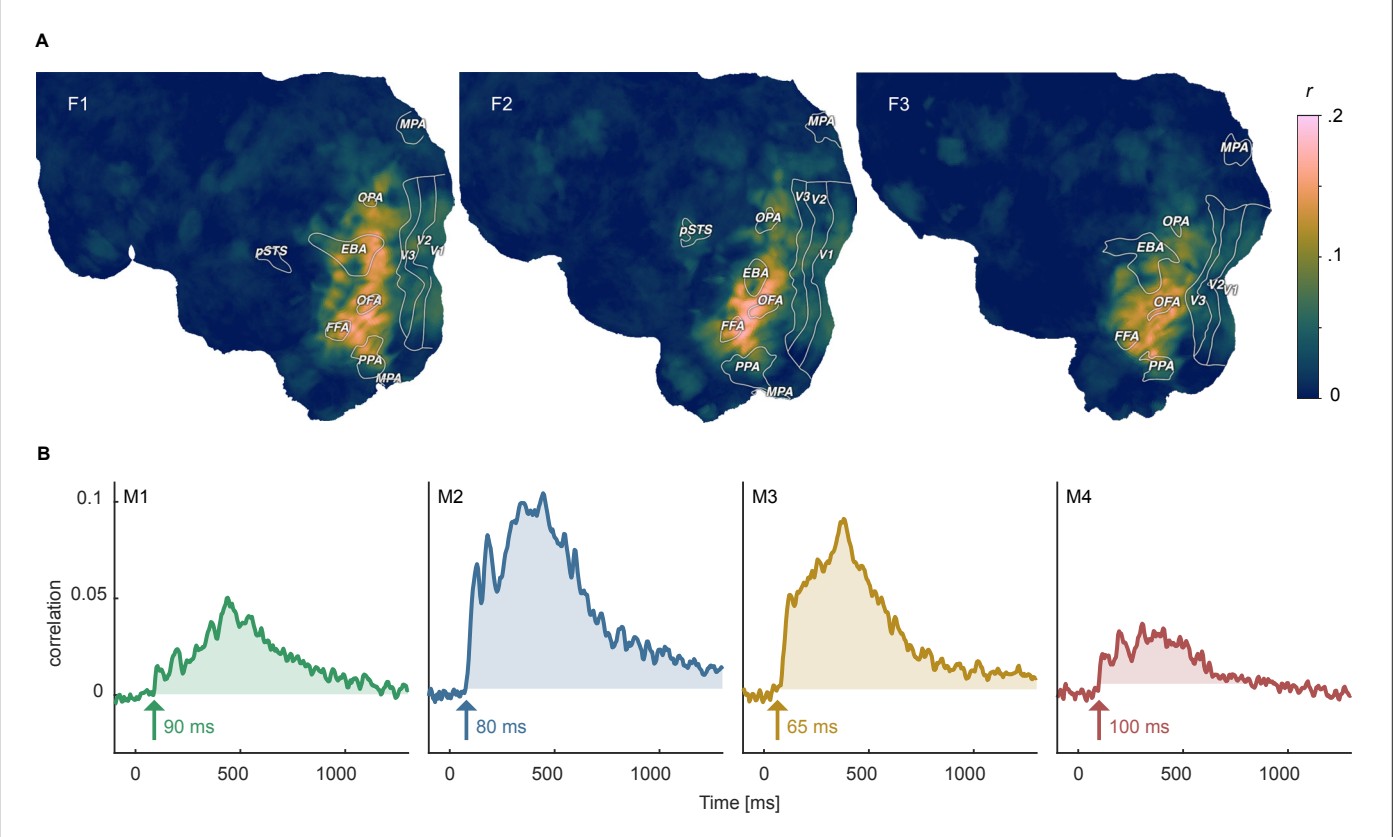

**Figure 7.** Identifying shared representations between brain and behavior. (**A**) Pearson correlation between perceived similarity in behavior and local fMRI activity patterns using searchlight representational similarity analysis. Similarity patterns are confined mostly to higher visual cortex. (**B**) Pearson correlation between the perceived similarity in behavioral and time-resolved MEG activation patterns across sensors using representational similarity analysis. The largest time window of timepoints exceeding a threshold are shaded. The threshold was defined as the maximum correlation found during the baseline period.

FFA response could only be predicted from later timepoints of the MEG signal (~180ms). This finding is in line with many studies showing face-specific effects measured with fMRI in FFA (*Kanwisher et al., 1997*; *Grill-Spector et al., 2004*; *Tong et al., 2000*) and a later dominance of high-level face responses (*Bentin et al., 1996*; *Deffke et al., 2007*; *Eimer, 2011*; *Wardle et al., 2020*). Contrasting the correlation time courses of V1 and FFA (*Figure 8B*), we found that the correlation with V1 was larger than that of FFA between 105 and 130ms. Together, these analyses highlight the potential for combining larger datasets to provide a detailed spatiotemporally-resolved account of object processing.

## Discussion

THINGS-data provides researchers in cognitive and computational neuroscience with a unique large-scale multimodal collection of neuroimaging and behavioral datasets in response to thousands of images of up to 1854 diverse objects. We have demonstrated the high quality of these datasets and we have provided five examples for potential research directions, including information-based multivariate decoding at the image and category level, data-driven visualization of response patterns across space and time, large-scale hypothesis testing by evaluating the reproducibility of previous research findings, revealing the relevance of the neuroimaging datasets for learning about behavioral similarity judgments, and regression-based fusion of MEG and fMRI data for uncovering a spatiotemporally resolved information flow in the human brain.

Two key strengths that set THINGS-data apart from other public datasets are its multimodality and size, offering fMRI and MEG responses to up to 22,448 object images collected over 12 sessions per participant and 4.70 million behavioral similarity judgments in response to natural object images,

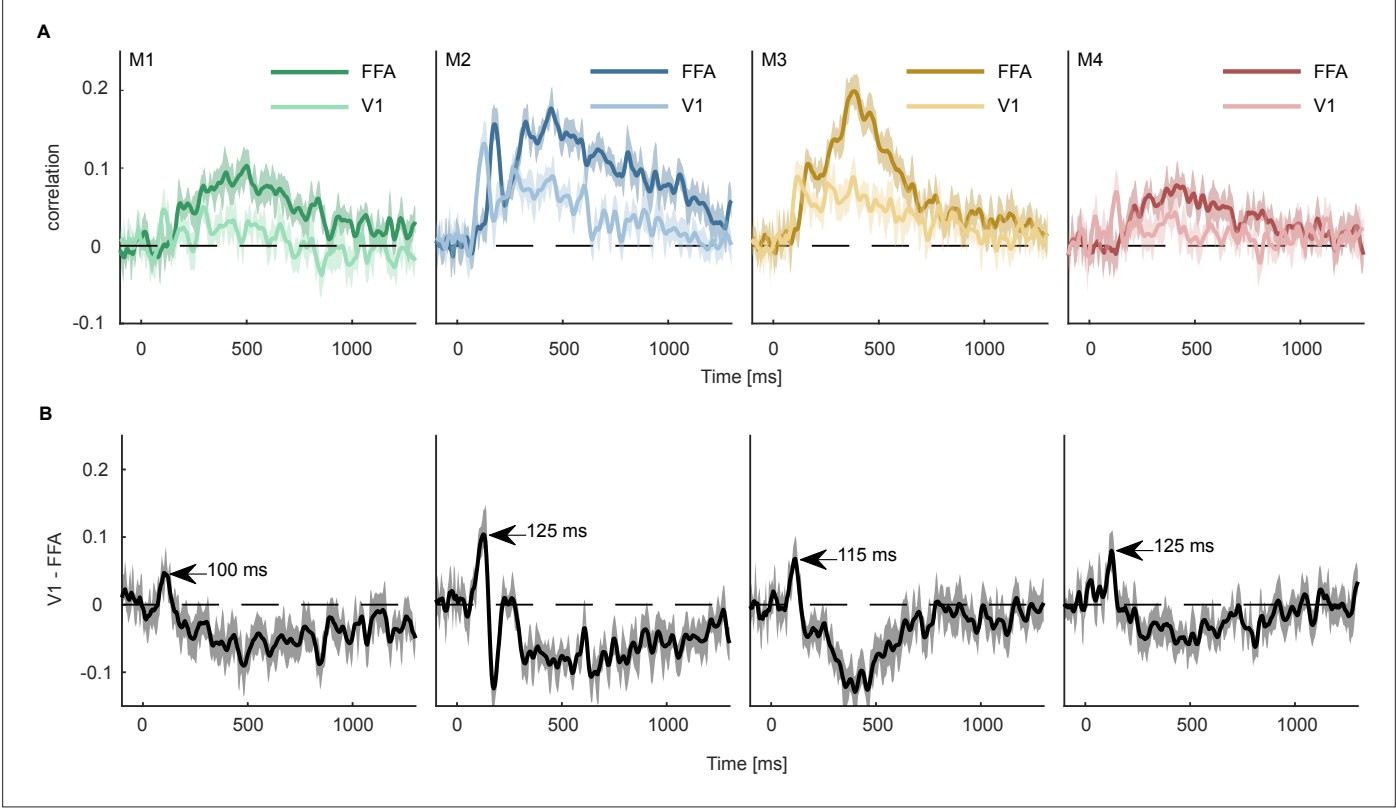

**Figure 8.** Predicting fMRI regional activity with MEG responses. (**A**) Pearson correlation between predicted and true regression labels using mean FFA and V1 responses as dependent and multivariate MEG sensor activation pattern as independent variable. Shaded areas around the mean show bootstrapped confidence intervals (n=10,000) based on a 12-fold cross-validation scheme, leaving one session out for testing in each iteration. The mean across the cross-validation iterations is smoothed over 5 timepoints. (**B**) Difference between V1 and FFA time-resolved correlations with the peak highlighting when the correlation with V1 is higher than that of FFA. Error bars reflect boostrapped 95% confidence intervals (10,000 iterations).

allowing countless new hypotheses to be tested at scale. For example, how are behaviorally relevant object dimensions reflected in patterns of brain activity, both in space and in time? What is the interplay of animacy, size, curvature, object color, object spikiness (*Bao et al., 2020*), and other dimensions for a broader set of natural objects (*Stoinski et al., 2022*)? How is object memorability represented in the human brain, and what other factors, such as object category or typicality, affect this representation (*Kramer et al., 2022*)? How stable are object representations across days, and how does this effect depend on object category? What are the differences in perceived object similarity across gender and age, both within category and between categories? By offering multiple datasets targeting semantically broad spatial and temporal brain responses as well as behavioral judgments in response to images, THINGS-data fills an important gap in our ability as researchers to bring together different data domains. Among others, our exemplary analyses demonstrate a new method for directly combining MEG and fMRI responses without the assumptions imposed by representational similarity analysis (*Cichy et al., 2014*; *Hebart and Baker, 2018*) or source modeling (*Baillet, 2017*).

Beyond the multimodality and size, by virtue of being based on the THINGS object concept and image database (*Hebart et al., 2019*), THINGS-data comes with a rich and growing source of meta-data specifically targeting cognitive and computational neuroscientists, including high-level categories, typicality, object feature ratings (*Stoinski et al., 2022*), as well as memorability scores for each individual image (*Kramer et al., 2022*). THINGS-data also provides numerous additional measures beyond the core datasets, including diverse structural and functional MRI data including resting state fMRI and functional localizers, physiological recordings for MRI data, and eye-tracking for MEG data. While eye-tracking data will be of limited use for studying natural eye movement dynamics, given the instruction of maintaining central fixation, these data can be used for targeted removal of MEG time periods involving occasional involuntary eye movements and eye blinks. Together with the large-scale

behavioral dataset, these extensive measures and their breadth promise key new insights into the cortical processing of objects in vision, semantics, and memory.

In line with a growing body of literature highlighting the strengths of densely sampled datasets (*Naselaris et al., 2021*; *Allen et al., 2022*; *Chang et al., 2019*; *Smith and Little, 2018*), for the fMRI and MEG datasets we targeted our efforts at extensive samples of a small number of participants instead of broadly sampling the population. Our key exemplary results replicate largely across participants, highlighting their generality and demonstrating that it is possible to treat each participant as a separate replication, with the aim of producing results that may generalize to the population. Another key benefit of extensively sampling individual brains is the ability to provide insights that generalize across objects and images (*Naselaris et al., 2021*). Our data quality assessments and exemplary results replicate a large number of existing research findings, demonstrating that THINGS-data yields replicable results and highlighting that, despite known issues in the replicability of neuroscience and behavioral research (*Marek et al., 2022*; *Button et al., 2013*; *Open Science Collaboration, 2015*), many previously reported findings are reliable and generalize across objects and images. At the same time, our exemplary analyses aimed at replicating previous fMRI and MEG work of size and animacy representation indeed only partially reproduced these findings (*Konkle and Caramazza, 2013*; *Konkle and Oliva, 2012*), highlighting the importance of extensive and representative sampling of object images. In order to confidently generalize results of specific hypotheses derived from THINGS-data to the population, additional focused studies in a larger set of participants may be conducted to strengthen these conclusions. THINGS-data thus offers an important testbed not only for new hypotheses but also for assessing the replicability and robustness of previous research findings.

The fMRI dataset published as part of THINGS-data provides important unique value beyond existing densely-sampled unimodal fMRI datasets targeting natural images. The present fMRI dataset contains responses to 720 objects and 8740 object images that were sampled to be representative of our visual diet (*Hebart et al., 2019*). In contrast, three other publicly available, large fMRI datasets of natural images (*Allen et al., 2022*; *Chang et al., 2019*; *Horikawa and Kamitani, 2017*) use images sampled from large machine learning databases, including Imagenet and MSCOCO (*Lin et al., 2014*; *Russakovsky et al., 2015*; *Xiao et al., 2010*), or are focused more strongly on natural scenes. While the advantage of these other datasets is the direct comparability with neural network models trained on these machine learning databases, this complicates the assessment of individual objects in scenes and comes with specific category biases that may affect the interpretation of results. For example, ImageNet contains a lot of dog images while lacking a person category, and MSCOCO is dominated by 80 categories (e.g. 'giraffe', 'toilet') that also often co-occur in images (e.g. 'dog' and 'frisbee'). Beyond selection bias, these existing fMRI datasets are limited in their coverage of object categories or provide only a single exemplar per category, which limits their utility for the study of object invariance specifically and object semantics more generally. One existing fMRI dataset provides individual images for a broad set of objects, yet without multiple exemplars per category (*Chang et al., 2019*), another dataset offers data with 8 object exemplars yet is restricted to 150 categories in the training set (*Horikawa and Kamitani, 2017*), and two datasets strongly sample the 80 MSCOCO categories (*Allen et al., 2022*; *Chang et al., 2019*). In contrast, THINGS-data offers individual objects from the THINGS database from 720 to 1854 carefully-curated object categories (*Hebart et al., 2019*), with 12 unique images per object. Finally, in THINGS-data, all images are presented only once to each participant, with the exception of test images, which precludes changes in brain responses across image repeats related to adaptation or memory, but which leads to lower reliability of BOLD estimates at the individual image level than slow event-related designs (*Chang et al., 2019*), block designs (*Horikawa and Kamitani, 2017*), or designs with repeated image presentations (*Allen et al., 2022*). Thus, while existing unimodal datasets may be particularly useful for comparing results to machine learning models, for exploratory data analyses or for modeling of natural scenes, it is unclear how well results from these previous datasets will generalize to the entire breadth of objects in our environment. In contrast, the fMRI dataset of THINGS-data offers a uniquely broad, comprehensive and balanced sampling of objects for investigating visual and semantic representations across the human brain.

Despite the semantic breadth of the datasets provided and the use of naturalistic images, there are important limitations of our datasets. First, while the datasets reflect a broad sampling strategy for object categories and object images, object categories in the THINGS database had been selected as being representative of the American English language, which may continue to yield residual biases in

the frequency of object categories that may lead to a biased assessment of object representations in brain and behavior. However, THINGS allows evaluating how robust findings are when changing the sampling strategy, which may help overcome this limitation. An additional limitation is the fact that our fMRI and MEG datasets relied on a small number of participants for making general statements about brain function. However, as discussed above, small-n designs have a long tradition in vision science and many of the studied effects replicate across participants, indicating that each participant can be treated as a replication of the same experiment (*Smith and Little, 2018*). Further, stimuli were chosen to be naturalistic images presented at fixation, yet our world does not consist of static snapshots of images viewed through a restricted frame but instead of a continuously moving world present at a much wider viewing angle. Due to the absence of well-controlled short naturalistic movies of objects, the technical limitations of presenting larger stimuli, and the added complexity of recording and analyzing datasets consisting of movies, we decided to rely on object images. Future studies may extend these efforts to naturalistic movies of objects embedded in scenes (*Huth et al., 2022*), yet with a similar level of semantic control as imposed by THINGS, and potentially in a wide angle environment, yielding important additional insights into object processing. Finally, any results may be affected by the choice of the task, which may have less of an effect on earlier processing stages yet more strongly affect object representations in anterior temporal lobe, as well as parietal prefrontal cortex (*Harel et al., 2014*; *Bracci et al., 2017*; *Hebart et al., 2018*). Future efforts should compare the effect of different tasks on object representations across cerebral cortex.

THINGS-data reflects the core release of the THINGS initiative (https://things-initiative.org), a global initiative bringing together researchers around the world for multimodal and multispecies collection of neuroimaging, electrophysiological, and behavioral datasets based on THINGS objects. As part of the THINGS initiative, two electroencephalography (EEG) datasets have recently been made available (*Gifford et al., 2022*; *Grootswagers et al., 2022*). In contrast to our temporally spaced MEG dataset that offers non-overlapping and unobstructed responses to stimuli, these datasets used a rapid serial visual presentation design, which allows presenting more images in a shorter time window, yet which leads to a strong overlap in neighboring responses and interactions between stimuli that are known to affect high-level processing (*Robinson et al., 2019*). While this and the improved spatial fidelity afforded by MEG promise significant unique value of our MEG dataset, the datasets that are available or will be made available as part of the THINGS initiative offer a unique opportunity for convergence across multiple methods, species and paradigms. In this context, THINGS-data lays the groundwork for understanding object representations in vision, semantics, and memory with unprecedented detail, promising strong advances for the cognitive and computational neurosciences.

## Methods
### Participants
For the onsite MRI and MEG studies, 3 healthy volunteers (2 female, 1 male, mean age at beginning of study: 25.33 years) took part in the MRI study and 4 different healthy volunteers (2 female, 2 male, mean age at beginning of study: 23.25 years) in the MEG study. Sample size of n=4 was determined in advance based on a trade-off between number of participants and effort of data collection. A fourth participant for the MRI study was planned but due to repeated technical issues and the ensuing lack of time was canceled. All on-site participants were screened prior to participation for suitability and availability, all with prior experience in studies that required keeping their eyes on fixation for prolonged periods of time. All participants were right-handed and had normal or corrected-to-normal visual acuity. Participants provided informed consent in participation and data sharing, and they received financial compensation for taking part in the respective studies. The research was approved by the NIH Institutional Review Board as part of the study protocol 93 M-0170 (NCT00001360).

For the online study, behavioral data were collected through the crowdsourcing platform Amazon Mechanical Turk. 14,025 workers participated in the triplet odd-one out experiment, for a total of 5,517,400 triplet choices. The sample size was determined based on the number of triplets expected to be sufficient for reaching threshold in data dimensionality, which was estimated to be ~5 million triplets. We collected an additional 10% to compensate for assumed partial exclusion of the data. A subset of 1.46 million triplets had been used in previous work (*Hebart et al., 2020*; *Zheng et al., 2019*; *Muttenthaler et al., 2022b*). Data quality was assessed separately across four batches.

Exclusion criteria were in part pre-established based on previous work (*Hebart et al., 2020*), in part updated to yield a decent trade-off between data quality and the amount of excluded data. Workers in a given batch were excluded if they were overly fast in at least five separate assignments of 20 trials each (>25% responses faster than 800ms and >50% responses faster than 1,100ms), overly repetitive in a set of ≥200 trials (deviating from the central 95% distribution), or very inconsistent in providing demographic information (>3 ages provided). These criteria led to the exclusion of 818,240 triplets (14.83%). The final dataset consisted of 12,340 workers (6619 female, 4400 male, 56 other, 1065 not reported; mean age: 36.71, std: 11.87, n=5170 no age reported) and 4,699,160 triplets, of which 4,574,059 triplets comprised the training and test data for computational modeling and 125,101 triplets the four datasets used for computing noise ceilings. Workers received financial compensation for their study participation ($0.10 for 20 trials, median RT per trial: 2846ms). Workers provided informed consent for the participation in the study. The online study was conducted in accordance with all relevant ethical regulations and approved by the NIH Office of Human Research Subject Protection (OHSRP).

## Stimuli

Images for all three datasets were taken from the THINGS object concept and image database (*Hebart et al., 2019*). THINGS comprises 1854 object concepts from a comprehensive list of nameable living and non-living objects and things, including non-countable substances (e.g. 'grass', 'sand'), faces (e.g. 'baby', 'boy', 'face'), as well as body and face parts (e.g. 'eye', 'leg'). For each concept, THINGS contains a minimum of 12 high-quality colored images of objects embedded in a natural background (total number of images: 26,107).

For the MEG dataset, all 1854 object concepts were sampled, with the first 12 exemplars per concept, for a total of 22,248 unique images presented once throughout the study. For the MRI dataset, given time limitation for the planned 12 sessions, sampling was restricted to a subset of 720 representative object concepts, again with the first 12 exemplars per concept, for a total of 8640 unique images (for the concept and image selection strategy, see Appendix 1). In addition, for the MEG dataset, there were 200 separate THINGS images that were among the remaining THINGS images. These images were presented repeatedly and served as a separate test set for model evaluation. For MRI, there were 100 separate test images that were a representative subset of the 200. Finally, there were 100 unique catch images that were created using the generative adversarial neural network BigGAN (*Brock et al., 2019*). These images were generated by interpolating between two latent vectors, yielding novel objects that were not recognizable. All presented images subtended 10 degrees of visual angle and were presented on a mid-grey background, and a fixation crosshair (*Thaler et al., 2013*) subtending 0.5 degrees was overlaid onto the image.

For the behavioral dataset, the 1854 images were used that had been shown during evaluation of the concepts included in the THINGS database (*Hebart et al., 2019*), of which 1663 images ended up overlapping with the THINGS images (other images had been excluded from the database because of small image size). The images were cropped to square size, with the exception of a small number of images for which objects didn't fit inside a square and which were padded with white background.

## Experimental procedure
### MRI study procedure

MRI participants wore custom fitted head casts (Caseforge Inc, USA) to minimize head motion and improve alignment between sessions. Stimuli were presented on a 32" BOLD screen (Cambridge Research Systems Ltd, UK) that was placed behind the bore of the scanner and viewed through a surface mirror attached to the head coil. Respiration and pulse were recorded at 500 Hz using a breathing belt and a photoplethysmograph, respectively (Biopac System Inc, USA).

Participants took part in a total of 15–16 scanning sessions. All sessions of a given participant took place roughly at the same time of day (+/-2 hours) to avoid non-specific effects associated with changes during the day (*Orban et al., 2020*; *Steel et al., 2019*). The first 1–2 sessions were used for testing the fit of the individualized head casts (see below) and for acquiring functional localizers for measuring retinotopic maps using population receptive field (pRF) mapping (4–6 runs, ~8 min each) as well as attaining category-selective functionally localized clusters in response to images of faces, body parts, scenes, words, and objects (6 runs, ~4.5 min each; for details, see Appendix 3). In the

next 12 sessions, functional data was acquired for the main experiment using THINGS images. In the last two sessions, two separate datasets were acquired that are planned to be published separately. During each session, if there was sufficient time, additional anatomical images were acquired (see MRI data acquisition). At the end of each session, a resting state run was conducted (~6 min, eyes closed).

Each of the 12 main fMRI sessions consisted of 10 functional runs (~7 min each). In each run, 72 object images were presented, as well as 10 test and 10 catch images. Participants' task was to keep their eyes on fixation and report the presence of a catch image with a button press on a fiber-optic diamond-shaped button box (Current Designs Inc, USA). Stimuli were presented for 500ms, followed by 4 s of fixation (SOA: 4.5 s). This amounted to a total of 92 trials per run, 920 trials per session, and 11,040 trials in total per participant. The 720 object images in a given session were chosen such that each of the 720 object concepts were present, while all 100 test images were shown in each session once and the catch images were chosen randomly. The order of trials was randomized within each functional run, with the constraint that the minimum distance between two subsequent catch images was three trials. Stimulus presentation was controlled using MATLAB with Psychtoolbox (*Brainard, 1997*; *Kleiner et al., 2007*).

## MEG study procedure

MEG participants wore an individually molded head cast (Chalk Studios Ltd, UK) to minimize head motion and improve alignment between sessions. Head position was measured with three marker coils attached to the head casts (nasion, as well as anterior to the left and right preauricular pits). Head position was recorded at the beginning and end of each run. Stimuli were presented on a back projection screen using a ProPixx projector (VPixx Technologies Inc, Canada). Eye position and pupil size was tracked at 1200 Hz throughout the study using an EyeLink 1000 Plus (SR Research, Canada).

Each MEG participant attended one MRI session and 14 MEG sessions. In the MRI session, a T1-weighted structural brain image (MPRAGE, 0.8 mm isotropic resolution, 208 sagittal slices) was collected without head padding to allow for the construction of a custom head cast and as part of the dataset to allow for improved MEG source modeling. The next 12 sessions were the main MEG sessions using THINGS images, while in the final two sessions, two separate datasets were acquired that are planned to be published separately. Within each of the 12 main sessions, the overall procedure was very similar to the MRI study, with the main difference that 1854 objects were presented in each session and that the stimulus presentation rate was faster. Each session consisted of 10 runs (~5 min each). In each run, 185–186 object images were presented, as well as 20 test and 20 catch images. Stimuli were presented for 500ms, followed by a variable fixation period of 1000±200ms (SOA: 1500±200ms). Jitter was included to reduce the effect of alpha synchronization with trial onset. This amounted to 225–226 trials per run, 2,254 trials per session, and 27,048 trials per participant. Stimulus presentation was controlled using MATLAB with Psychtoolbox (*Brainard, 1997*; *Kleiner et al., 2007*).

## Online crowdsourcing study procedure

The triplet odd-one out task was collected using the online crowdsourcing platform Amazon Mechanical Turk. The task was carried out in a browser window. On a given trial, participants saw three images of objects side by side and were asked to indicate with a mouse click which object they perceived as the odd-one out. Then, the next trial was initiated after 500ms. To reduce bias, participants were told to focus on the object but no additional instructions were provided as to what constitutes the odd-one out. Each task consisted of 20 trials, and workers could choose to participate as often as they liked. This had the advantage that workers could stop whenever they no longer felt motivated to continue. After completing the 20 trials, workers were prompted to fill in demographic information. For the first set of ~1.46 million trials, workers could voluntarily report gender and ethnicity, while for the remaining dataset, workers could voluntarily report gender, ethnicity, and also age. Triplets and stimulus order were chosen randomly, but were selected in a way that each cell of the final 1,854×1,854 similarity matrix was sampled roughly equally often. In the final dataset, each cell was sampled on average 7.99 times, with all cells sampled at least once and 98.48% of all cells sampled 6 times or more. For a small subset of 40,000 trials, participants were shown the same set of 1,000 triplets twice within the same task (i.e. 40 per triplet), with a minimum distance of 16 trials to reduce short-term memory effects. The purpose of this manipulation was to estimate an upper bound for the consistency

of participants' choices. For another subset of 40,000 trials this same set of triplets was shown but this time to different participants, to estimate the lower bound for the consistency of participants' choices. Finally, two other subsets of trials were generated with two other sets of 1000 triplets (25,000 and 40,000 trials, respectively), to ensure that data quality remained stable across data acquisition time periods. Stimulus presentation was controlled with custom HTML and Javascript code.

## MRI acquisition and preprocessing

### MRI data acquisition

All magnetic resonance images were collected at the NIH in Bethesda, MD (USA) using a 3 Tesla Siemens Magnetom Prisma scanner and a 32-channel head coil. During the main task of the fMRI experiment involving the THINGS images, we collected whole-brain functional MRI data with 2 mm isotropic resolution (60 axial slices, 2 mm slice thickness, no slice gap, matrix size 96×96, FOV = 192 × 192 mm, TR = 1.5 s, TE = 33ms, flip angle = 75°, echo spacing 0.55ms, bandwidth 2,264 Hz/pixel, multi-band slice acceleration factor 3, phase encoding posterior-to-anterior).

We collected additional high-resolution data of each participant's individual anatomy (2–3 T1-weighted and one T2-weighted images per participant), vasculature (Time-of-Flight and T2*-weighed), and functional connectivity (resting state functional data), as well as gradient echo field maps to account for image distortions due to inhomogeneities in the magnetic field. The resting state functional MRI data was acquired using the reverse phase encoding direction (anterior-to-posterior) compared to the main functional runs to allow for an alternative method for distortion correction (*Andersson et al., 2003*). A detailed description of the MRI imaging parameters can be found in *Supplementary file 1*.

### MRI data preprocessing

Functional magnetic resonance imaging data was deidentified (*Gulban et al., 2019*), converted to the Brain Imaging Data Structure (*Gorgolewski et al., 2016*) and preprocessed with fMRIPrep (*Esteban et al., 2019*) (version 20.2.0). A detailed description of this procedure can be found in the online dataset on figshare (see Data Availability). In short, the preprocessing pipeline for the functional images included slice timing correction, rigid body head motion correction, correction of susceptibility distortions based on the field maps, spatial alignment to each participant's T1-weighted anatomical reference images, and brain tissue segmentation and reconstruction of pial and white matter surfaces. Since the default pipeline of fMRIPrep does not allow the inclusion of multiple T1-weighted and T2-weighted anatomical images, which can improve each participant's surface reconstruction and all downstream processing steps, we manually ran Freesurfer's recon-all (*Dale et al., 1999*) and passed the output to fMRIPrep. Finally, we visually inspected the cortical surface reconstruction and manually placed relaxation cuts along anatomical landmarks including the calcarine sulcus to generate cortical flat maps for visualization purposes (*Gao et al., 2015*). Preprocessing and analysis of retinotopic mapping data yielded retinotopic visual regions V1-V3, hV4, VO1/VO2, LO1/LO2, TO1/TO2, and V3a/V3b (see Appendix 3). Preprocessing and analysis of functional localizer data yielded fusiform face area (FFA), occipital face area (OFA), posterior superior temporal sulcus (pSTS), extrastriate body area (EBA), parahippocampal place area (PPA), medial place area / retrosplenial complex (MPA), occipital place area (OPA), and lateral occipital cortex (LOC). For subject F3, pSTS could not be defined because no significant cluster of face-selective activation was localized in that area.

### fMRI ICA denoising

fMRI data contains noise due to head motion, pulse and heartbeat, respiration, as well as other physiological and scanner-related factors that can negatively impact downstream data analysis (*Murphy et al., 2013*). Independent component analysis (ICA) has been shown to reliably separate many signal and noise components (*Beckmann and Smith, 2004*). However, common existing automatic or semi-autom atic ICA classification approaches are based either on a complex classification pipeline (*Salimi-Khorshidi et al., 2014*) which may be prone to overfitting, or they are focused on head motion-related artifacts alone (*Pruim et al., 2015*). Therefore, we developed a heuristic semi-automated classification approach to capture a broad set of physiological and scanner-related artifacts based on independent component analysis (ICA).

For attaining stable independent components, each functional run was additionally preprocessed with spatial smoothing (FWHM = 4 mm) and a high-pass filter (cut-off=120 s). Decomposing the preprocessed data of each run with MELODIC ICA (*Beckmann and Smith, 2004*) yielded a total of 20,389 independent components for all sessions of all 3 participants. For each independent component, we quantified a set of features which we hypothesized to be related to its potential classification as signal or noise, which are explained in more detail below: The correlation with the experimental design, the correlation with physiological parameters, the correlation with head motion parameters, its edge fraction, and its high-frequency content.

The correlation with the experimental design was estimated by convolving the stimulus onsets with a canonical hemodynamic response function and computing the Pearson correlation with the component time series. The correlation with physiological parameters was taken as the maximum correlation of the component time series with a set of physiological regressors derived from the raw cardiac and respiratory recordings (see code make_physio_regressors.m). Similarly, the correlation with head motion was taken as the maximum correlation of the component time series with any of the head motion estimates produced by fMRIPrep. The edge fraction reflects the presence of high independent component weights near the edge of the brain and was estimated as the sum of absolute weights in the edge mask, divided by the sum of absolute weights within the entire brain mask. The edge mask was generated by subtracting an eroded brain mask (eroded by 4 mm) from the original whole-brain mask. High-frequency content was defined as the frequency at which the higher frequencies explain 50% of the total power between 0.01 Hz and the Nyquist frequency (*Pruim et al., 2015*).

Once these features had been derived, two independent raters manually labeled a subset of all independent components. We randomly sampled a set of 2472 components (1614 for rater 1; 1665 for rater 2; 807 of which were rated by both). Raters gave each component a unique label for either signal, head motion noise, physiological noise, MR-scanner noise, other noise source, or unknown, as well as a confidence rating from 1 (not confident) to 3 (very confident). Interrater agreement for labeling components as signal vs. not signal was 87%. Raters made their choices based on summary visualizations (*Figure 2—figure supplement 5*), which showed each component's respective spatial map, time series, and temporal frequency spectrum as well as additional information including (1) the time course of the experimental design, (2) the expected time course of physiological noise and (3) the expected time course of head motion related noise. The time course of the experimental design was created by convolving the stimulus onsets with a canonical HRF. We estimated the expected time course of physiological noise by regressing the physiological confounds against the component time series and visualized the resulting prediction. Similarly, we estimated the expected time course of head motion related noise by regressing head motion parameters against the component time course and visualized the resulting prediction. The head motion parameters used included rotation and translation along the three axes, as well as their square value and first derivative. Finally, these visualizations also showed the highest Pearson correlation of the component time series with the physiological confounds and the head motion parameters as well as the correlation with the experimental design, the high frequency content and the edge fraction.

We then visually inspected the distributions of the labeled data along the estimated feature dimensions. The results showed that the signal distribution was reliably separable from the noise distributions based on edge fraction and high-frequency content alone. For robustness, we defined univariate thresholds in these features (edge fraction: 0.225, high-frequency content: 0.4) and classified each of the 20,388 originally estimated components accordingly (rater 1: 61% noise sensitivity, 98% signal specificity; rater 2: 69% noise sensitivity, 98% signal specificity). The resulting noise component time series were then used as noise regressors for the single trial response estimation in downstream analyses. Incorporating these noise regressors strongly improved the reliability of single trial response estimates (*Figure 1—figure supplement 1*).

## fMRI single-trial response estimates

Beyond the many sources of noise in fMRI data, another challenge lies in the fact that fMRI voxel-wise time series consist of a lot of data, making analyses of the entire dataset computationally challenging and potentially inaccessible to researchers with fewer resources. The time series format is also not ideal for some data-driven techniques such as multivariate pattern analysis (*Haxby et al., 2001*), representational similarity analysis (*Kriegeskorte et al., 2008a*), or dimensionality reduction

techniques that require discrete data samples as inputs. To overcome these challenges, we estimated the BOLD response amplitude to each object image by fitting a single-trial general linear model on the preprocessed fMRI time series. Our procedure was similar to the recently-developed GLMsingle approach (*Allen et al., 2022*; *Prince et al., 2022*), but we adopted an approach to better suit (1) our experimental design which contained image repeats only across sessions and (2) the use of ICA noise regressors which varied in number between runs.

First, we converted data from each functional run to percent signal change. We then regressed the resulting time series data against a set of noise regressors comprising the ICA noise components for that run and a set of polynomial regressors up to degree 4. The residuals of this step were then kept for downstream analyses. Although a stepwise regression approach can be suboptimal (*Lindquist et al., 2019*), we chose it in order to avoid regularizing the noise regressors when optimizing the single trial beta estimates, and post-hoc analyses demonstrated that resulting noise ceilings were, indeed, slightly higher with our approach. To account for differences in the shape of the hemodynamic response function (HRF), we used a library of 20 HRFs that had previously been shown to capture a broad range of HRF shapes (*Allen et al., 2022*; *Prince et al., 2022*) and determined the best fitting HRF for each voxel. To this end, we generated a separate on-off design matrix for each of the 20 HRFs, fit each design matrix to the fMRI time series separately, and determined the best HRF per voxel by the largest amount of explained variance. Since the regressors of neighboring trials are highly correlated in a fast event-related design, we used fractional ridge regression to mitigate overfitting and to identify the regularization parameter for each voxel that maximized the predictive performance of left-out data (*Rokem and Kay, 2020*). We used a range of regularization parameters from 0.1 to 0.9 in steps of 0.1 and from 0.9 to 1.0 in steps of 0.01 to sample the hyperparameter space more finely for values which correspond to less regularization. We evaluated the performance based on the consistency of beta estimates over repeatedly presented trials in a leave-one-session-out cross-validation. To this end, we determined the sum of squared differences between the mean of the regularized betas in the 11 training sessions and the unregularized betas in the held-out session. We then fit a single-trial model with the best hyperparameter combination per voxel (HRF and fractional ridge parameter) to obtain the set of single-trial beta coefficients. Since ridge regression leads to biases in the overall scale of the beta coefficients, we linearly rescaled them by regressing the regularized against the unregularized coefficients and keeping the predictions as the final single-trial response amplitudes.

The resulting beta weights of the single-trial model represent an estimate of the BOLD response amplitude to each object image as a single number per voxel. This data format is much smaller than the original time series, is amenable to a wider range of analysis techniques, and was used for all analyses showcased in this manuscript. Both the voxel-wise time series and single-trial response estimates are publicly available such that users may choose the data format that best suits their research purposes (see Data availability).

## MEG acquisition and preprocessing

### MEG data acquisition

The MEG data were recorded with a CTF 275 MEG system (CTF Systems, Canada) which incorporates a whole-head array of 275 radial 1st order gradiometer/SQUID channels. The MEG was located inside a magnetically shielded room (Vacuumschmelze, Germany). Data were recorded at 1,200 Hz. 3rd gradient balancing was used to remove background noise online. Recordings were carried out in a seated position. Since three MEG channels were dysfunctional (MLF25, MRF43, and MRO13), data were available from 272 channels only. Eye-tracking data (position and pupil) were saved as signals in miscellaneous MEG channels (x-coordinate: UADC009, y-coordinate: UADC010, pupil size: UADC013). Parallel port triggers were used to mark the stimulus onset in real time (channel: UPPT001). To account for temporal delays between the computer and the stimulus display, we used an optical sensor which detects light changes (channel: UADC016).

### MEG data preprocessing and cleaning

We used MNE-python (*Gramfort et al., 2013*) to preprocess the MEG data. We bandpass filtered the raw data for each run between 0.1 and 40 Hz. For two participants (M1 and M2), there was a complete MEG signal dropout in one run which lasted for less than 200ms. We replaced this segment of the data with the median sensor response before epoching. To mark trial onsets in the continuous

MEG recordings, we used parallel port triggers and the signal of an optical sensor which detects light changes on the display and can thus account for temporal delays between the computer and the projector. We used the signal from the optical sensor to epoch the continuous data from –100ms to 1300ms relative to stimulus onset. We then baseline corrected the epoched data by subtracting the mean and dividing by the standard deviation of the data during baseline (100ms before stimulus onset).Next, we excluded one sensor (MRO11) for all participants which was regularly unlocked and thus yielded very noisy responses. After all preprocessing steps were completed, data were downsampled to 200 Hz to reduce computational load for downstream analyses.

## MEG head motion

Continuous head localization did not allow for stable MEG recordings, so it was deactivated. However, given the use of individualized head casts, we expected head motion to be minimal during runs. We recorded snapshots of head position via the three marker coils before and after every run. To examine how much of a concern head motion is, we calculated the within- and the cross-session changes in head position for every participant using the data from the average of the pre- and post-run measurements. We found that within-session head motion was minimal (median <1.5 mm for all participants), with slightly larger but still small head motion across sessions (median <3 mm for all participants). Note that differences in measured head position could also be due to the positioning of the marker coils inside the head cast. For one participant (M4), it seems likely that the marker coils in two sessions were not positioned in the exact same location as in other sessions (see *Figure 2—figure supplement 4*).

## MEG event-related fields

To examine the stability of evoked responses across sessions, we inspected the event-related fields for the 200 test images that were repeated in each session. As the 200 images showed different objects, we expected a stable visual response across sessions. Thus, we averaged the response across all 200 test images for occipital, temporal, and parietal sensors, respectively, and plotted the evoked responses for each sensor and sensor group separately (see *Figure 2—figure supplement 1*). Overall, the visualization indicates that the evoked responses for each participant were similar across the twelve sessions, highlighting that differences between sessions (e.g. in head motion or cognitive state) were likely not detrimental to the overall results.

## Noise ceiling estimation

We estimated noise ceilings for all datasets as an indicator of data reliability. The noise ceiling is defined as the maximum performance any model can be expected to achieve in terms of explainable variance (*Lage-Castellanos et al., 2019*). For the fMRI and MEG data, we estimated the noise ceiling based on the variability of responses to the subset of object images which were presented repeatedly in each session (fMRI: 100 images, 12 repetitions; MEG: 200 images, 12 repetitions). To this end, we used the analytical approach introduced recently (*Allen et al., 2022*). The noise variance was estimated as the pooled variance over repeated responses to the same image. The signal variance was estimated by first taking the mean response over repetitions of the same image and then computing the variance over the resulting image-wise average responses. Finally, the total variance was taken as the sum of the noise and signal variance. Thus, the noise ceiling was defined as the ratio between the signal variance and the total variance. This expression of the noise ceiling can be adjusted to account for the number of trials researchers might want to average in their analysis (*Allen et al., 2022*). We computed two noise ceiling estimates: one for the test set (adjusted for averaging over 12 trial repeats) and one for single trial responses (unadjusted). Note that the noise ceiling was estimated independently for each measurement channel (i.e. each voxel in fMRI and each timepoint and sensor in MEG).

For the behavioral dataset, there were three noise ceiling datasets where triplets were sampled repeatedly between participants, and one where they were sampled within participants. For the three between-subject noise ceiling datasets, a different set of 1000 random triplets were chosen, while the within-subject noise ceiling triplets were the same as the second dataset. Several noise ceiling datasets were acquired to test for non-stationarity in the acquired behavioral dataset, since the first 1.46 million triplets had been acquired much earlier than the later datasets. For a given triplet, across

all participants that had taken part, the choice consistency was computed in percent. The noise ceiling was then defined as the average choice consistency across all triplets. Note that this procedure slightly overestimates the true noise ceiling since it is always based on the most consistent choice in the sample.

## Behavioral modeling procedure

The procedure for deriving a low-dimensional embedding from triplet odd-one out judgments has been described in detail previously (*Hebart et al., 2020*). The computational model is available online (https://github.com/ViCCo-Group/SPoSE; *Muttenthaler et al., 2022a*), which was implemented in PyTorch 1.6 (*Paszke et al., 2019*). First, we split up the triplets into a training and test set, using a 90–10 split. The SPoSE model is designed to identify a set of sparse, non-negative and interpretable dimensions underlying similarity judgments. The embedding was first initialized with 90 random dimensions (range 0–1). Next, for a given triplet of images, the dot product of the 90-dimensional embedding vectors between all three pairs of images was computed, followed by a softmax function (no temperature parameter), yielding three predicted choice probabilities. The highest choice probability was then used as a basis for the computation of the loss. The loss function for updating the embedding weights consisted of the cross-entropy, which is the logarithm of the softmax function, and a separate sparsity-inducing L-1 norm of the weights, where the trade-off between both loss terms is determined by a regularization parameter $\lambda$ that was identified with cross-validation on the training set (final $\lambda$: 0.00385). Optimization was carried out using Adam (*Kingma and Ba, 2017*) with default parameters and minibatch size of 100 triplets. Once optimization had completed, all dimensions with weights exclusively <0.1 were removed. We then sorted the dimensions in descending order based on the sum of the weights.

Since the optimization procedure is stochastic, the resulting embedding will change slightly depending on how it is initialized. To identify a highly reproducible embedding, we ran the entire procedure 72 times with a different random seed, yielding 72 separate embeddings. For a given embedding and a given dimension, we then iterated across the remaining 71 embeddings, identified the most similar dimension, and used this to compute a reproducibility index (average Fisher-z transformed Pearson correlation). Repeating this process for each dimension in an embedding provided us with a mean reproducibility for each embedding. We then picked the embedding with the best reproducibility, yielding the final embedding with 66 dimensions. One of the authors (MNH) then visually inspected and hand labeled all 66 dimensions. Note that the same author had generated the labels for all 49 dimensions in the original model, which mostly agreed with participants' labels to these dimensions (*Hebart et al., 2020*).

## Extrapolation from small dataset to predict saturation of dimensionality

While it has been shown previously that the modeling procedure approached peak performance already with a smaller dataset (*Hebart et al., 2020*), the embedding dimensionality kept increasing, indicating a benefit of collecting a larger dataset for a more refined representational embedding. To determine how large a dataset was required until model dimensionality no longer grew noticeably, we estimated the growth in model dimensionality as a function of dataset size by extrapolating the original dataset of 1.46 million trials. To achieve this aim, we first took the estimated dimensionality of 4 embedding that had been computed at each step of 100,000 trials up until 1.4 million trials, making it a total of 14 steps and 56 embeddings. Next, we fitted an exponential decay function with the shape of $a + b\,e^{-cx}$ to the mean dimensionality across all 14 steps and extrapolated this function. Finally, we computed 1000 bootstrap estimates by resampling the means from all 4 embeddings per position and repeating this fitting procedure. This was used to identify 95% confidence intervals of the estimated model dimensionality given the dataset size. In the limit, the embedding saturated at 67.54 dimensions (95% CI: 61.94–74.82). The final dataset size was determined as a trade-off between approaching the final model dimensionality and data acquisition cost.

## Fine-grained prediction of perceived similarity

To identify the degree to which the updated embedding yielded improved prediction of fine-grained similarity, we used 8 existing datasets from three studies (*Avery et al., 2022*; *Iordan et al., 2022*;

*Peterson et al., 2018*) that had examined within category similarity. Note that predicted similarities are likely underestimated, given that the original similarity datasets were collected using different image examples and/or tasks. First, we took the labels from these datasets and identified the overlap with the THINGS concepts, while adjusting small differences (e.g. 'chairs' was changed to 'chair', 'mandrill' to 'monkey'). Several datasets contained multiple images per object concept, i.e. not all used concepts were unique. This yielded datasets of the high-level categories animal (n=10, all concepts unique concepts, Iordan; n=104, 39 unique, Peterson), food (n=30, all unique, Avery), fruit (n=72, 24 unique, Peterson), furniture (n=81, 12 unique, Peterson), vegetable (n=69, 23 unique, Peterson), and vehicle (2 datasets, n=10, all unique, Iordan; n=78, 13 unique, Peterson). Since the SPoSE model allows for computing similarity within a constrained context, we used category-constrained similarity estimates, using all examples of a given superordinate category to generate similarity estimates for a given model. The representational similarity was then computed using the lower triangular part of each matrix and using Pearson correlation between the similarity matrices derived from the original 49-dimensional embedding and the measured similarity matrix, as well as the new 66-dimensional embedding and the measured similarity matrix. Finally, we computed 100,000 bootstrap estimates for each representational similarity to attain confidence estimates, by repeatedly sampling rows and columns in each similarity matrix referring to different individual objects. To test if across all eight datasets there was an overall improvement, we determined the fraction of bootstrap examples yielding a mean improvement in predicted similarity. When five or more similarity matrices showed an improvement, this was counted as an improvement (>50%), while if 4 or fewer similarity matrices showed an improvement, this was counted as no improvement (≤50%). The fraction of cases where there was no improvement was then taken as the p-value.

## fMRI and MEG multivariate decoding analyses

To validate the usefulness of the neuroimaging datasets for studying object representations, we conducted two sets of multivariate decoding analyses (*Hebart and Baker, 2018*; *Haynes, 2015*) focused at revealing object-related information content in brain activity patterns. One set of analyses was conducted at the level of object images, while the other was carried out at the level of object concepts. The object image analyses were based on the 100 test images for fMRI and the 200 test images for MEG, respectively, of which each had been repeated 12 times each. The object concept analyses were based on all 12 unique exemplars per object concept that had not been repeated (fMRI: 720 concepts; MEG: 1,854 concepts). All analyses were conducted using leave-one-session-out cross-validation. For fMRI, we used spatially-resolved searchlight decoding on the beta weights (radius = 10 mm), implemented in The Decoding Toolbox (*Hebart et al., 2014*), while for MEG, we used time-resolved decoding at the sensor level implemented in the CoSMoMVPA toolbox (*Oosterhof et al., 2016*). FMRI analyses were based on pairwise linear support vector machine classification (258,840 pairwise classifiers per searchlight and cross-validation step) using default hyperparameters in LIBSVM (*Chang and Lin, 2011*), while MEG analyses were based on pairwise linear discriminant analysis (1,717,731 pairwise classifiers per timepoint and cross-validation step) with default hyperparameters. Iteratively training classifiers on 11 sessions and testing them on the 12th session yielded representational dissimilarity matrices based on classification accuracy for all pairs of comparisons. The reported accuracies reflect the mean of the lower triangular part of these matrices, which corresponds to the mean pairwise decoding accuracy (chance: 50%).

## fMRI and MEG multidimensional scaling

To explore the representational structure in fMRI and MEG response patterns evoked by different objects, we visualized their relationships using multidimensional scaling (MDS). For demonstrating the utility of the dataset for identifying meaningful structure from patterns of brain activity alone, we specifically focused on the spatial clustering of known superordinate category information in the datasets. For fMRI, we extracted image-specific parameter estimates from lateral occipital complex and all previously defined category-selective ROIs, with the exception of medial place area, and averaged them across exemplars for each object concept, yielding 720 voxel response patterns at the object concept level. We then fit 2D-MDS based on the correlation distance between these response patterns (10 initializations, 5000 iterations, implemented in scikit-learn *Pedregosa et al., 2011*). For MEG, we directly used the time-resolved pairwise decoding accuracy matrices for all 1854 object concepts

from the previous analysis step, fit 2D-MDS in a time-resolved fashion, and iteratively aligned results across time using Procrustes transformation (implemented in the functions cmdscale and procrustes in MATLAB). For plotting the resulting two-dimensional embeddings (*Figure 5B and E*), we highlighted superordinate categories with different colors, and for MEG we visualized equally spaced time points with 200ms distance.

## Object animacy and size analyses

We aimed at identifying the degree to which previously reported neuroimaging findings regarding object animacy and size generalize to our larger neuroimaging datasets. To this end, we used human animacy and size ratings for all 1854 object concepts, obtained as part of the extended THINGS+ metadata (*Stoinski et al., 2022*). In short, animacy ratings for each object concept in the THINGS database (*Hebart et al., 2019*) were collected by presenting raters with the respective noun and asking them to respond to the property 'something that lives' on a Likert scale. Real-world size ratings for each object concept were obtained in two steps. First, raters were instructed to indicate the size of a given object noun on a continuous scale, defined by nine reference objects spanning the size range of all objects (from 'grain of sand' to 'aircraft carrier'). In each trial, raters first indicated the approximate size. In a second step, the rating scale zoomed in on the interval between the closest two anchor points in order to allow raters to give a more refined answer.

For fMRI, we first fit a simple ordinary least squares linear regression model to the average fMRI response for each object concept (smoothed with FWHM = 3 mm), using z-scored ratings as predictors. Then, we visualized the voxel-wise regression weights on the cortical surface as indicators for the preferred tuning to animate vs. inanimate and big vs. small objects, respectively. For MEG, we ran time-resolved cross-validated ordinary least squares linear regression predicting size and animacy ratings from MEG sensor activation patterns. Note that the direction of inference here is reversed as compared to fMRI for better comparability to previous research findings. Cross-validation was implemented in a leave-one-session-out fashion (11 training sessions, 1 test session) and was based on the correlation between the predicted and the true animacy and size ratings.

## Multimodal analyses

### Relating neuroimaging data to behavioral similarity judgments

To demonstrate a use case for integrating the behavioral dataset with the neuroimaging datasets, we conducted representational similarity analysis (*Kriegeskorte et al., 2008a*), comparing representational dissimilarity matrices from patterns of fMRI voxel activity and MEG sensor activity with those obtained for behavioral similarity judgments. To this end, we first computed a large-scale behavioral similarity matrix for all 1854 objects, where object similarity for a given pair of objects $i$ and $j$ was defined as the triplet choice probability for choosing object $k$ as the odd-one out, averaged across all 1852 possible $k$, which was estimated from the choices predicted from the 66-dimensional SPoSE embedding. Next, we converted this matrix to a dissimilarity matrix and extracted its lower triangular part, separately for the 720 concepts for fMRI and all 1854 concepts for MEG. We then took the existing pairwise decoding matrices for all fMRI searchlights and MEG time points that had been computed for the pairwise decoding analyses at the object concept level (see *fMRI and MEG multivariate decoding analyses*), extracted their lower triangular part, and compared it to the behavioral similarity matrix using Pearson's correlation. This resulted in a representational similarity estimate for each fMRI searchlight location and MEG time point, highlighting the spatial and temporal distribution of effects related to perceived similarity of objects.

### Regression-based MEG-fMRI fusion

We aimed at demonstrating the usefulness of integrating the multimodal neuroimaging datasets for revealing insights into spatio-temporal evolution of object-related information in the human brain. To this end, the sheer size of the datasets allowed us to combine MEG and fMRI data directly using multiple linear regression (regression-based MEG-fMRI fusion). For our demonstration, we focused on two regions of interest, V1 and FFA, and used the MEG data to predict the univariate BOLD response in these regions. First, we averaged the responses in V1 and FFA across all three fMRI participants. Next, for every timepoint separately, we trained an ordinary least squares linear regression model on MEG sensor data from 11 sessions to predict the response for each trial in V1 and FFA. Then, we

used the parameter estimates to predict the fMRI response using the left-out MEG data. We then correlated the predicted V1/FFA response with the true V1/FFA response to examine at what time-points the image-specific effects observed in V1 and FFA emerged in the MEG data.

## Code availability

Code for implementing the main neuroimaging analyses described in this manuscript is available on GitHub (https://github.com/ViCCo-Group/THINGS-data; *Contier et al., 2023*). All relevant code for reproducing results of the behavioral dataset can be found on OSF (https://osf.io/f5rn6/, https://doi.org/10.17605/OSF.IO/F5RN6).

## Acknowledgements

We thank Elissa Aminoff, Kendrick Kay, Alex Martin, Thomas Naselaris, Francisco Pereira, and Michael Tarr for useful discussions in the design stage of these datasets. Additional thanks to Tom Holroyd, Sean Marrett, Frank Sutak, and Dardo Tomasi for technical support with the fMRI and MEG facilities and to Govind Bhagavatheeshwaran and Sean Marrett for support designing MRI sequences. Thanks to James Gao for continued support with generating and using custom MRI head cases. Special thanks to Ed Silson for sharing the functional and retinotopic localizer code, to Christian Büchel for allowing us to use and share their code for converting raw physiological recordings to physiological regressors, and to Lukas Muttenthaler for creating a faster and more versatile version of the SPoSE embedding code. Thanks to Ülkühan Tonbuloglu and Julia Norman for manual labeling of independent compo-nents. We are grateful for useful discussions with Kendrick Kay and Jacob Prince on single trial param-eter estimates and with Talia Konkle on object animacy and size effects. We would like to thank Jason Avery, Marius Cătălin Iordan, and Joshua Peterson for sharing their object similarity matrices. To run the MEG analysis we utilized the computational resources of the NIH HPC Biowulf cluster. (http://hpc.nih.gov). We are grateful to Jeff Stout, Ashita Basavaraj, and Anthony Galassi for their help with using the HPC. This work was supported by the Intramural Research Program of the National Institutes of Health (ZIA-MH-002909, ZIC-MH002968), under National Institute of Mental Health Clinical Study Protocol 93 M-1070 (NCT00001360), a research group grant by the Max Planck Society awarded to MNH, the ERC Starting Grant project COREDIM (101039712), and the Hessian Ministry of Higher Education, Science, Research and Art (LOEWE Start Professorship to MNH and Excellence Program "The Adaptive Mind"). The funders had no role in study design, data collection and analysis, decision to publish or preparation of the manuscript.

## Additional information

### Competing interests

Chris I Baker: Senior editor, *eLife*. The other authors declare that no competing interests exist.

### Funding

| Funder | Grant reference number | Author |
| --- | --- | --- |
| National Institutes of Health | ZIA-MH-002909 | Chris I Baker |
| National Institutes of Health | ZIC-MH002968 | Charles Y Zheng |
| Max-Planck-Gesellschaft | Max Planck Research Group M.TN.A.NEPF0009 | Martin N Hebart |
| European Research Council | Starting Grant StG-2021-101039712 | Martin N Hebart |
| Hessisches Ministerium für Wissenschaft und Kunst | LOEWE Start Professorship | Martin N Hebart |
| Max Planck School of Cognition | | Oliver Contier |

| Funder | Grant reference number | Author |
|---|---|---|
| Hessisches Ministerium für Wissenschaft und Kunst | Tha Adaptive Mind | Martin N Hebart |

The funders had no role in study design, data collection and interpretation, or the decision to submit the work for publication. Open access funding provided by Max Planck Society.

## Author contributions

Martin N Hebart, Conceptualization, Resources, Data curation, Software, Formal analysis, Validation, Investigation, Visualization, Methodology, Writing – original draft, Project administration, Writing – review and editing; Oliver Contier, Data curation, Software, Formal analysis, Validation, Visualization, Methodology, Writing – original draft, Project administration, Writing – review and editing; Lina Teichmann, Data curation, Software, Formal analysis, Validation, Visualization, Methodology, Writing – original draft, Writing – review and editing; Adam H Rockter, Data curation, Investigation, Writing – review and editing; Charles Y Zheng, Software, Writing – review and editing; Alexis Kidder, Anna Corriveau, Maryam Vaziri-Pashkam, Investigation, Writing – review and editing; Chris I Baker, Conceptualization, Supervision, Funding acquisition, Writing – original draft, Writing – review and editing

## Author ORCIDs

Martin N Hebart http://orcid.org/0000-0001-7257-428X
Oliver Contier http://orcid.org/0000-0002-2983-4709
Lina Teichmann http://orcid.org/0000-0002-8040-5686
Adam H Rockter http://orcid.org/0000-0002-2446-717X
Maryam Vaziri-Pashkam http://orcid.org/0000-0003-1830-2501
Chris I Baker http://orcid.org/0000-0001-6861-8964

## Ethics

Clinical trial registration NCT00001360.
Human subjects: All research participants for the fMRI and MEG studies provided informed consent in participation and data sharing, and they received financial compensation for taking part in the respective studies. The research was approved by the NIH Institutional Review Board as part of the study protocol 93-M-0170 (NCT00001360). All research participants taking part in the online behavioral study provided informed consent for the participation in the study. The online study was conducted in accordance with all relevant ethical regulations and approved by the NIH Office of Human Research Subject Protection (OHSRP).

## Decision letter and Author response

Decision letter https://doi.org/10.7554/eLife.82580.sa1
Author response https://doi.org/10.7554/eLife.82580.sa2

# Additional files

## Supplementary files

• Supplementary file 1. Acquisition parameters for all MRI sequences used in the fMRI dataset.
• MDAR checklist

## Data availability

All parts of the THINGS-data collection are freely available on scientific data repositories. We provide the raw MRI (https://doi.org/10.18112/openneuro.ds004192.v1.0.5) and raw MEG (https://doi.org/10.18112/openneuro.ds004212.v2.0.0) datasets in BIDS format (*Gorgolewski et al., 2016*) on Open-Neuro (*Markiewicz et al., 2021*). In addition to these raw datasets, we provide the raw and preprocessed MEG data as well as the raw and derivative MRI data on Figshare (*Thelwall and Kousha, 2016*) at https://doi.org/10.25452/figshare.plus.c.6161151. The MEG data derivatives include preprocessed and epoched data that are compatible with MNE-python and CoSMoMVPA in MATLAB. The MRI data derivatives include single trial response estimates, category-selective and retinotopic regions of interest, cortical flatmaps, independent component based noise regressors, voxel-wise noise ceilings, and estimates of subject specific retinotopic parameters. In addition, we included the

preprocessed and epoched eyetracking data that were recorded during the MEG experiment in the OpenNeuro repository. The behavioral triplet odd-one-out dataset can be accessed on OSF (https://osf.io/f5rn6/).

The following datasets were generated:

| Author(s) | Year | Dataset title | Dataset URL | Database and Identifier |
|---|---|---|---|---|
| Hebart MN, Contier O, Teichmann L, Rockter AH, Zheng CY, Kidder A, Corriveau A, Vaziri-Pashkam M, Baker CI | 2022 | THINGS-fMRI | https://doi.org/10.18112/openneuro.ds004192.v1.0.5 | OpenNeuro, 10.18112/openneuro.ds004192.v1.0.5 |
| Hebart MN, Contier O, Teichmann L, Rockter AH, Zheng CY, Kidder A, Corriveau A, Vaziri-Pashkam M, Baker CI | 2023 | THINGS-MEG | https://doi.org/10.18112/openneuro.ds004212.v2.0.0 | OpenNeuro, 10.18112/openneuro.ds004212.v2.0.0 |
| Hebart MN, Contier O, Teichmann L, Rockter AH, Zheng CY, Kidder A, Corriveau A, Vaziri-Pashkam M, Baker CI | 2022 | THINGS-odd-one-out | https://doi.org/10.17605/OSF.IO/F5RN6 | Open Science Framework, 10.17605/OSF.IO/F5RN6 |
| Hebart MN, Contier O, Teichmann L, Rockter A, Zheng C, Kidder A, Corriveau A, Vaziri-Pashkam M, Baker C | 2023 | THINGS-data: A multimodal collection of large-scale datasets for investigating object representations in brain and behavior | https://doi.org/10.25452/figshare.plus.c.6161151.v1 | Figshare, 10.25452/figshare.plus.c.6161151.v1 |

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

## Appendix 1

### Concept and image selection strategies

The 720 categories as well as the representative test sets of 100 and 200 images were selected based on two criteria: to maximize overlap with the concepts included in the machine learning image dataset Ecoset (*Mehrer et al., 2021*) and to be visually and conceptually as representative of the overall THINGS image set as possible. Ecoset offers a more natural distribution of object concepts than typically used machine learning image databases and has a strong overlap in concepts with those used in THINGS. Maximal overlap with Ecoset was thus chosen to allow for better training of neural networks using the same concepts and thus better comparability with THINGS-data. To select images that are visually and conceptually representative of the overall image set, we first selected the intersection of concepts between Ecoset and THINGS and included these concepts (n=470). Next, we ran spectral clustering (k=80) on all THINGS concepts and images using activations separately in two layers of the brain-inspired neural network CorNet (*Kubilius et al., 2019*): Layer V1 and layer IT, with the aim of being representative of early and high-level visual processing. Finally, we additionally ran spectral clustering (k=80) on the 49 dimensions from the original computational model derived from behavioral odd-one-out choices (*Hebart et al., 2020*), with the aim of being representative of mental representations of objects in humans. For the selection of the subsets of 720 concepts, we next identified the concepts that were as representative as possible of all 240 clusters and their cluster sizes, using a greedy selection approach iteratively swapping in and out pairs of images until all clusters were sampled representatively. Once the 720 categories had been determined, we repeated this approach for the 200 images for the MEG dataset, this time based on all remaining images of the 720 concepts that are not used as main experimental stimuli. Finally, from these 200 images, we selected the 100 most representative for the MRI dataset.

## Appendix 2

### Eye-tracking processing and results

During the MEG sessions, we recorded eye-tracking data using an EyeLink 1000 Plus Eye-Tracking System as a means for controlling that participants were able to maintain central fixation. Eye movements were recorded from one eye, with x-coordinates, y- coordinates, and pupil size being fed directly into miscellaneous sensors of the MEG (sensors UADC009-2104, UADC010-2101, UADC013-2104, respectively) with a sampling rate of 1200 Hz. We preprocessed the continuous eye-tracking data before epoching in the same way as the MEG data (−100–1300ms relative to stimulus onset) to assess how well participants fixated.

### Eye-tracking preprocessing

We preprocessed the eye-tracking data separately for each run (*Appendix 2—figure 1A*) and based our pipeline on previous work (*Allen et al., 2022*; *Kret and Sjak-Shie, 2019*). First, we removed invalid samples which we defined as x- and y- eye positions beyond the stimulus edges (10°). Then, we removed samples based on pupil dilation speed to detect eyeblinks. We calculated the pupil dilation changes from one timepoint to the next and examined when the dilation speed changed more than a threshold. The threshold was determined by examining the median absolute deviation from all dilation speeds in the run multiplied by a constant of 16 (*Kret and Sjak-Shie, 2019*). As the dilation speed threshold may not detect initial phases of the eyelid closure, we expanded the gap by 100ms before and 150ms after the blink occurred (*Allen et al., 2022*). We then removed samples that were temporally isolated (≥40ms away from other valid measurements) and only had a few consecutive valid measurements around them (max. 100ms). In addition, we fitted a smooth line to the pupil size data and excluded samples with a larger deviation (*Kret and Sjak-Shie, 2019*). Finally, we ran linear detrending on the x- and y-coordinates as well as the pupil size data to account for slow drifts over the run. On average, we removed ~10% of the eye-tracking samples during preprocessing (*Appendix 2—figure 1B*).

### Eye-tracking results

The results (*Appendix 2—figure 1C*) show that all four MEG participants fixated well. The gaze position of all participants was within 1 degree of the stimulus in more than 95% of all valid samples (*Appendix 2—figure 1C*). Looking at the time-resolved data, it seems that on average there was only minimal time-locked eye movement (max. 0.3 degrees, see *Appendix 2—figure 1D*). In addition, we found no consistent pattern of pupil size changes across time (see *Appendix 2—figure 1E*). Together, this indicates that participants mostly fixated during the MEG experiment.

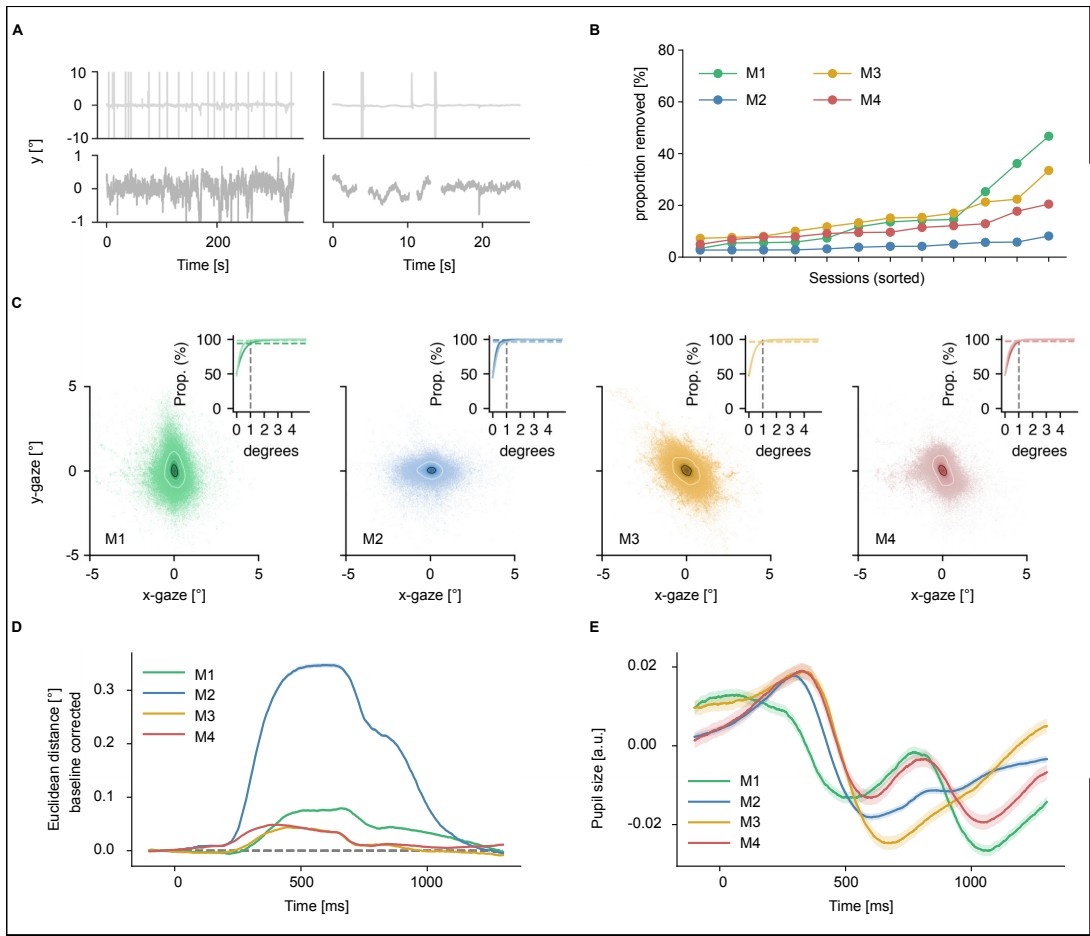

**Appendix 2—figure 1.** Eye-tracking preprocessing and results. (**A**) Visual illustration of the eye-tracking preprocessing routine. Raw data for one example run (top row) and preprocessed data for the same run (bottom row). (**B**) Amount of eye-tracking data removed during preprocessing in each session for each participant separately, sorted by proportion removed. On average we lost around 10% of the eye-tracking samples during preprocessing. (**C**) Gaze positions for all four participants. The large panel shows eye positions across sessions for each participant (downsampled to 100 Hz). To quantify fixations, we added rings to the gaze position plots corresponding to containing 25% (black) and 75% (white) of the data. In addition, we examined the proportion of data falling below different thresholds (small panel top right corner within the large panels). The vertical dashed lines indicate the 1 degree mark in all panels. (**D**) Mean time-resolved distance in gaze position relative to the baseline period in each trial. Shading represents standard error across all trials. (**E**) Time-resolved pupil size in volts. Larger numbers reflect a larger pupil area. Shading represents standard error across sessions.

# Appendix 3

## fMRI population receptive field mapping and category localizer

### fMRI population receptive field mapping and early visual regions of interest

The purpose of the pRF experiment was to estimate subject-specific retinotopy by stimulating different parts of the visual field. We adapted a paradigm used in previous studies (*Silson et al., 2015*). Participants saw natural scene images through a bar aperture that swept across the screen. Stimuli were presented on a mid-grey background masked by a circular region (10.6° radius). Bars swept along 8 directions (horizontal, vertical, and diagonal axes, bidirectional). Each bar sweep was split into 18 positions, each lasting 3 s (54 s per sweep), and 10 scene stimuli were presented briefly (300ms) within the mask at each position. Each of the 90 scene images were presented twice in each sweep. A functional run (~8 min) entailed the bar mask sweeping along all 8 directions, plus an additional 15 s of fixation in the beginning and end. Participants carried out a task at fixation where they had to indicate a change in color of the white fixation dot. Participants performed 4–6 functional pRF runs during the localizer sessions.

We estimated each subject's individual retinotopy based on a population receptive field model (*Dumoulin and Wandell, 2008*) of the sweeping bar experiment. As additional preprocessing, we applied a temporal filter to each functional run (100 s high pass) and normalized the resulting voxel-wise time series to zero mean and unit variance before averaging them across functional runs. We estimated retinotopic parameters - eccentricity, polar angle, and receptive field size - in each voxel based on a circular population receptive field model as implemented in AFNI (*Cox, 1996*). After projecting these results to the cortical surface, we used a Bayesian mapping approach that further refined these individual parameter estimates based on a predefined prior retinotopic group atlas that automatically delineates retinotopic visual regions accordingly (*Benson and Winawer, 2018*). These retinotopic visual regions of interest include V1-V3, hV4, VO1/VO2, LO1/LO2, TO1/TO2, and V3a/V3b, which were also resampled from the individual subject's cortical surface representation to functional volume space.

### fMRI localizer of object category selective regions

The aim of the category localizer experiment was to identify brain regions responding selectively to specific object categories. To this end, we adapted a functional localizer paradigm used in previous studies (*Groen et al., 2021*). Participants saw images of faces, body parts, scenes, words, objects, and scrambled object images in a block design. Each category block was presented twice per functional run with a duration of 15 s. Each functional run also contained fixation periods of 15 s in the beginning and 30 s in the end (4.5 min in total). The experiment included four functional runs, and the order of blocks within each run was randomized.

We aimed at identifying brain regions that are known to show increased activity to images of specific object categories. To this end, we fitted a general linear model (GLM) to the fMRI data of the object category localizer experiment (FSL version 5.0 *Woolrich et al., 2001* as implemented in Nipype *Gorgolewski et al., 2011*). Each functional run was spatially smoothed with a FWHM of 5 mm and entered in a GLM with regressors for body parts, faces, objects, scenes, words, and scrambled objects. We defined T-contrasts to estimate the selective response to object categories (body parts >objects, faces >objects, scenes >objects, objects >scrambled). The resulting statistical parametric maps were aggregated across functional runs within each subject with a fixed effects model (*Woolrich et al., 2004*) and corrected for multiple comparisons (cluster p-threshold=0.0001, extent-threshold=3.7). The resulting subject-specific clusters were intersected with an existing group parcellation of category-selective brain areas (*Julian et al., 2012*) to yield the final regions of interest: Fusiform face area (FFA), occipital face area (OFA), posterior superior temporal sulcus (pSTS), extrastriate body area (EBA), parahippocampal place area (PPA), medial place area / retrosplenial complex (MPA), occipital place area (OPA), and lateral occipital cortex (LOC).

