## [Editor Report]

Hebart et al., present a landmark, multimodal massive dataset to support the study of visual object representation, including data measured from functional magnetic resonance imaging, magnetoencephalography, and behavioral similarity judgments. The compelling, condition-rich design, conducted over a thoughtfully curated and sampled set of object concepts will be highly valuable to the cognitive/computational/neuroscience community, yielding data that will be amenable to many empirical questions beyond the field of visual object recognition. The dataset is accompanied by quality control evaluations, as well as examples of analyses that the community can re-run and further explore for building new hypotheses that can be tested with such a rich dataset.

---

## [Decision Letter]

**Decision letter after peer review:**

Thank you for submitting your article "THINGS-data: A multimodal collection of large-scale datasets for investigating object representations in brain and behavior" for consideration by *eLife*. Your article has been reviewed by 3 peer reviewers, and the evaluation has been overseen by a Reviewing Editor and Floris de Lange as the Senior Editor. The following individuals involved in the review of your submission have agreed to reveal their identity: Talia Konkle (Reviewer #2); Enrico Glerean (Reviewer #3).

Essential revisions:

In our consultation session, we were all in agreement that this work is excellent. Each reviewer offers a slightly different perspective, and you can see their full feedback in context below. In general, the revisions we view as essential are as follows:

1. Providing more clarity to the open code, within what is possible given your skillset (i.e., we are not asking you to become software engineers).

2. We thought that the paper might be more impactful if some of the potential applications were highlighted and described in more detail (per Reviewer 1). We wish to emphasize that no new analyses are required unless you wish to conduct them.

3. Providing additional clarity on some of the methods, as requested by the reviewers.

In addition, each reviewer offered some additional comments, which we hope you will consider as you revise the manuscript.

*Reviewer #1 (Recommendations for the authors):*

My feedback below does not affect the authors' main conclusions and I see all these as addressable with a revision.

1. One important strength of this work is that it reveals fMRI datasets with few participants (n = 4) that can be replicable and reliable. Given recent discussions related to replication issues for certain types of studies (e.g., brain-wide associations), I wonder if the authors may wish to highlight the replicability of their work more directly. For example, it may be useful to define "exemplary analysis" (e.g., mentioned on pg 16) in the introduction and then describe that nearly a dozen neuroimaging results were replicated in the author's approach.

2. An additional paragraph in the discussion describing the strengths and weaknesses of THINGS relative to existing datasets may be needed. For example, the authors report analyses focused on ventral visual stream regions. Do the authors think that their dataset can be used to study other brain regions such as the prefrontal cortex or anterior temporal lobes that may be also involved in object recognition?

3. In general, I thought the authors could have been clearer in terms of the potential new insights this dataset can offer the field. For example, the authors mention "…THINGS-data will serve as an important resource for the community, enabling novel analyses to provide significant insights into visual object processing…" – pg. 3, and a series of research directions focused on methodology, e.g.: "including information-based multivariate decoding at the image and category level, data-driven visualization of response patterns across space and time, large-scale hypothesis testing by evaluating the reproducibility of previous research findings, revealing the relevance of the neuroimaging datasets for learning about behavioral similarity judgments, and regression-based fusion of MEG and fMRI data for uncovering a spatiotemporally-resolved information flow in the human brain as validation and extension of existing findings.." – pg. 15.

To clearly showcase the importance and impact of THINGS-data, I wonder if it may be needed to conduct an additional exploratory analysis that is not conditional upon a previous replication or alternatively, speculate in the Discussion the potential theoretical insights that may be gleaned from future work using THINGS.

4. Eye-tracking data was collected and reflects an additional source of behavioral data with which to relate to neural measures. Yet the authors mention this data only once in the introduction and once in the methods and do not report any results other than briefly in the Supplemental. Perhaps related to the above, I wished to see more elaboration, either through additional analysis or a discussion on how eye-tracking data can be used in a future study.

5. The authors suggest that correlating RDMs between MEG and fMRI is indirect and introduces additional assumptions (pg. 14). As these assumptions are never described, I am left unclear with how and what exactly about the authors approach can account for this potential problem.

6. Why does the number of dimensions capturing human similarity judgments increase with dataset size in Figure 3, especially related to the author's previous work (e.g., the original 49 dimensions from THINGS)?

*Reviewer #2 (Recommendations for the authors):*

For the behavioral measures, I wasn't entirely sure I followed some of the statements you made-e.g. you estimated a saturation at 4.5M trials, but wouldn't you have to run more trials to confirm this saturation effect? Also, the idea that you might not need more data to get changes in embedding dimensionality seemed slightly at odds with the subsequent points that participant-specific modeling is more reliable and lead to new insights (are their participant-specific dimensions? Not entirely sure how these were related, and also to within-category triads). Importantly, for me, I think these points are not key to the value of the behavioral similarity measures for the THINGs dataset. Really for me the key results are about data reliability-and you have it, and that these images are the same as the neural data so can now be linked to brain/meg data… wide open frontiers here. For me, exactly how it relates to and extends your prior work is interesting but almost in the same way as the other demonstrative analyses you conducted… there is more work to do to dig deeper into this. (And that's exciting!).

Style comments related to the narrative arc of the results. TOTALLY OPTIONAL, feel free to do none of these. No need to even reply.

– For me, the level of the method detail early in the Results sections was a bit too deep (e.g. describing the ICA denoising procedures and ridge regression approach; talking about the "edge fraction and high-frequency content"). For me, this level of detail in the results detracted just a little bit from the flow, because I didn't quite understand it and I had an easier time when I read the extended methods, and saw Sup Figure 10. Similarly, you report head motion parameters for the brain datasets in the main text and figures, but I also think these graphs could also be relegated to the Supplement. I found results like Sup Figure 2--better reliability after denoising--to be the key data I was looking for regarding your choices.

The 66-dimensional embedding of the behavioral Results section comes in between fmri/meg data reliability and fmri/meg decoding. I think you were going for reliability/data quality measures for all 3 datasets, then results of richer analyses. But I'm wondering if maybe you can think of decoding quality as part of data quality, and keep those fMRI/EEG sections together more? Or, maybe signpost this Results section organization more?

*Reviewer #3 (Recommendations for the authors):*

I do not have major concerns to raise for this manuscript, however, I strongly believe that the work presented would be even more valuable if the code attached to the manuscript could be improved for clarity and follow "good enough" standards for research software (https://journals.plos.org/ploscompbiol/article?id=10.1371/journal.pcbi.1005510). I know that software is not yet a valuable academic output compared to manuscripts, but I believe that the impact that will have on the community – along with the shared data – will be beyond the results and descriptions of this manuscript. In the list of comments below, I outline a minimal set of improvements to make the software more reusable. I am aware that researchers are not expected to be software engineers and I do understand if some of the requests might be too difficult, so please do as much as you can according to your skills.

List of improvements for the attached software

1. Licenses

Licenses are important because they let future users know if they can re-use your code. There are two zip files for the code in your manuscript. One contains licenses of the used packages and license for your code, the other did not mention any license. Please add at least a license for your code. If you are unsure, I recommend Apache 2.0 https://www.apache.org/licenses/LICENSE-2.0

2. README

Currently, there is a readme in PDF in one zip file describing the MEG scripts. There is no readme for the fmri part. There is no readme for the other zip file. Please include a REAME file in each of the two repositories (a text file is better than PDF, in general use PDF only for figures. Any text can be a text file). The readme file should list all files included in the repository and a short explanation of what they do

3. Folder structure

right now there is a mixture of scripts/results/other files to prepare containers/other files from other packages all in the same folder. Please consider separating the files into meaningful subfolders. There are no rules beyond the "good enough practices" paper linked above. At least separating results/derivative files from the code would be helpful.

4. Dependencies:

there are no dependencies listed. You need to specify which version of Matlab was used, you need to add the environment.yaml file for the conda environment that you activate in one script (if you are unsure please rune "conda env export > env.yml" after activating that environment). You need to confirm that all python scripts were run with that environment (conda activate is only present in the reconall script). You need to include the version of docker you used and the version of neurodocker. Since docker and/or neurodocker might change in the future, it would be recommended to also add the docker image you obtained to the repository or push it to dockerhub for people to reuse your docker images without needing to build them.

5. Testing and how to run

You need to document how a user can run and test the scripts themselves. For example, I have noticed that the script assumes that the BIDS (?) data should be located at a few parent subfolders. You could reconsider simplifying the work of the users, create a subfolder "BIDS" and tell the user to put there the data from neurovault and make your scripts point to the BIDS subfolder.

6. Accepting contributions and improvements from the community + version control

This is totally optional: right now it is difficult for a member of the community to recommend changes to your code because your code is not stored in a version control system (github, gitlab, etc). Please consider storing the code on github (or gitlab) and engage with your community of users by encouraging them to improve your code, and add future analysis to the same repository. In the long run, your repository (along with the dataset) could be a very valuable resource for the community, especially if you start getting contributions and code from future reuses of the data. Furthermore, by using a git repository you will also have the added benefits of version control of your software. Unlike manuscripts, the software is dynamic and so are the data, you can keep on improving some of your analysis or functions or just make the code more reusable by other scientists and the changes made will be documented automatically by the git system.

---

## [Author Response]

Essential revisions:In our consultation session, we were all in agreement that this work is excellent. Each reviewer offers a slightly different perspective, and you can see their full feedback in context below. In general, the revisions we view as essential are as follows:1. Providing more clarity to the open code, within what is possible given your skillset (i.e., we are not asking you to become software engineers).2. We thought that the paper might be more impactful if some of the potential applications were highlighted and described in more detail (per Reviewer 1). We wish to emphasize that no new analyses are required unless you wish to conduct them.3. Providing additional clarity on some of the methods, as requested by the reviewers.In addition, each reviewer offered some additional comments, which we hope you will consider as you revise the manuscript.

We would like to thank the editors and reviewers for highlighting these key points. We have addressed each of them below by (1) clarifying the openly available code, the software versions used, and by publishing and improving all code on GitHub, (2) highlighting additional potential applications of these data – in contrast to existing data and in contrast to small-scale datasets, and (3) clarifying methodological details by moving non-critical sections to Appendices and adding more detail in the main text where necessary.

Responses to each of the specific points raised by the reviewers are below.

Reviewer #1 (Recommendations for the authors):My feedback below does not affect the authors' main conclusions and I see all these as addressable with a revision.1. One important strength of this work is that it reveals fMRI datasets with few participants (n = 4) that can be replicable and reliable. Given recent discussions related to replication issues for certain types of studies (e.g., brain-wide associations), I wonder if the authors may wish to highlight the replicability of their work more directly. For example, it may be useful to define "exemplary analysis" (e.g., mentioned on pg 16) in the introduction and then describe that nearly a dozen neuroimaging results were replicated in the author's approach.

We would like to thank the reviewer for this excellent suggestion. In the revised manuscript, we now better highlight both reliability and replicability: (1) the degree of reliability between participants, which is highly relevant given the small sample size; and (2) replicability of other research, highlighting the value of our dataset and demonstrating its usefulness for testing replicability of existing research more broadly. As suggested, we now also define “exemplary analysis” in the introduction and highlight the number of studies replicated using our approach in the Discussion.

2. An additional paragraph in the discussion describing the strengths and weaknesses of THINGS relative to existing datasets may be needed. For example, the authors report analyses focused on ventral visual stream regions. Do the authors think that their dataset can be used to study other brain regions such as the prefrontal cortex or anterior temporal lobes that may be also involved in object recognition?

We agree with the reviewer. We have expanded the section in the discussion comparing the neuroimaging datasets of THINGS-data to other datasets and – also in response to Reviewer #3 – highlight the strengths and weaknesses of THINGS more broadly. We additionally discuss the potential utility of these datasets beyond ventral visual cortex in a new limitations paragraph in the Discussion section.

3. In general, I thought the authors could have been clearer in terms of the potential new insights this dataset can offer the field. For example, the authors mention "…THINGS-data will serve as an important resource for the community, enabling novel analyses to provide significant insights into visual object processing…" – pg. 3, and a series of research directions focused on methodology, e.g.: "including information-based multivariate decoding at the image and category level, data-driven visualization of response patterns across space and time, large-scale hypothesis testing by evaluating the reproducibility of previous research findings, revealing the relevance of the neuroimaging datasets for learning about behavioral similarity judgments, and regression-based fusion of MEG and fMRI data for uncovering a spatiotemporally-resolved information flow in the human brain as validation and extension of existing findings.." – pg. 15.To clearly showcase the importance and impact of THINGS-data, I wonder if it may be needed to conduct an additional exploratory analysis that is not conditional upon a previous replication or alternatively, speculate in the Discussion the potential theoretical insights that may be gleaned from future work using THINGS.

We would like to thank the reviewer. We believe this to be one of the most critical points raised by the reviewer. While we considered conducting additional exploratory analyses, we intentionally avoided overly novel and possibly controversial findings which might distract from the main message of releasing the three datasets for others to conduct new analyses. That said, we agree that it makes sense to better discuss the theoretical insights that these datasets may provide over and above small-scale datasets, and we now cover this more extensively in the Discussion section.

4. Eye-tracking data was collected and reflects an additional source of behavioral data with which to relate to neural measures. Yet the authors mention this data only once in the introduction and once in the methods and do not report any results other than briefly in the Supplemental. Perhaps related to the above, I wished to see more elaboration, either through additional analysis or a discussion on how eye-tracking data can be used in a future study.

We thank the reviewer for highlighting the eye-tracking data. Please note that participants were instructed to maintain central fixation throughout, which we confirmed empirically. While fixations were not perfect, this precludes an unbiased assessment of eye position. We now mention this restriction for the use of eye-tracking data more specifically. In addition, we mention a potential use case for the data in post-hoc cleaning of data (e.g. removal of time periods with eye blinks).

5. The authors suggest that correlating RDMs between MEG and fMRI is indirect and introduces additional assumptions (pg. 14). As these assumptions are never described, I am left unclear with how and what exactly about the authors approach can account for this potential problem.

We now better explain the assumptions introduced by RSA-based MEG-fMRI fusion in the main text. Hopefully, this makes the advantage of the approach we present clearer.

6. Why does the number of dimensions capturing human similarity judgments increase with dataset size in Figure 3, especially related to the author's previous work (e.g., the original 49 dimensions from THINGS)?

In our previous work8, we highlighted that the accuracy of our embedding for predicting triplet odd-one out choices had already saturated (Extended Data Figure 4a). However, we also discussed the limitation that the resulting dimensionality was a function of dataset size (Extended Data Figure 4b) and that highly similar dimensions appeared to be merged (e.g. plant-related / green). By extrapolating the dimensionality of the original dataset, before we collected additional data, we predicted that the number of dimensions would saturate around 66 to 67, with a dataset size of ~5 million triplets. This is very much in line with what we now find empirically. This indicates that while the 49 dimensions were already providing an excellent description of the representational space, the 66 dimensions reported here offered a more fine-grained view for which collecting additional data would be expected to yield only a minor improvement, which may no longer justify the cost of acquiring additional triplets. We now better explain this line of reasoning in the main text.

Reviewer #2 (Recommendations for the authors):For the behavioral measures, I wasn't entirely sure I followed some of the statements you made-e.g. you estimated a saturation at 4.5M trials, but wouldn't you have to run more trials to confirm this saturation effect? Also, the idea that you might not need more data to get changes in embedding dimensionality seemed slightly at odds with the subsequent points that participant-specific modeling is more reliable and lead to new insights (are their participant-specific dimensions? Not entirely sure how these were related, and also to within-category triads). Importantly, for me, I think these points are not key to the value of the behavioral similarity measures for the THINGs dataset. Really for me the key results are about data reliability-and you have it, and that these images are the same as the neural data so can now be linked to brain/meg data… wide open frontiers here. For me, exactly how it relates to and extends your prior work is interesting but almost in the same way as the other demonstrative analyses you conducted… there is more work to do to dig deeper into this. (And that's exciting!).

We thank the reviewer for pointing this out, and we fully agree that these are not key results. Yet, we thought it was important to explain why we collected a larger set of triplets, why we stopped at 4.5 million, and what would be the value in this larger dataset. We believe we could have described the motivation for our analyses more clearly. Our intention was not to measure a saturation but to demonstrate that our prediction about a saturation effect prior to data collection led to a dimensionality that we observed empirically. Given that the prediction was fulfilled, we assume that adding more data would not have a strong effect on embedding dimensionality, or at least the benefit may not justify the cost. In addition, the statements about whether additional data collection may be justified only referred to *global* similarity at the *population* level. However, there definitely is room for improvement for *within-category* (e.g. animal, vehicle) similarity estimates and similarity measurements at the *individual-participant* level for getting at interindividual differences. The former would benefit from a more targeted sampling, while the latter would require a different sampling approach (many triplets within individuals are difficult to collect). We now better explain this motivation for the analyses in the revised manuscript and highlight the potential refinements that could be made in future work. We also clarify the nature of the within-subject noise ceilings in the updated text.

Style comments related to the narrative arc of the results.– For me, the level of the method detail early in the Results sections was a bit too deep (e.g. describing the ICA denoising procedures and ridge regression approach; talking about the "edge fraction and high-frequency content"). For me, this level of detail in the results detracted just a little bit from the flow, because I didn't quite understand it and I had an easier time when I read the extended methods, and saw Sup Figure 10. Similarly, you report head motion parameters for the brain datasets in the main text and figures, but I also think these graphs could also be relegated to the Supplement. I found results like Sup Figure 2--better reliability after denoising--to be the key data I was looking for regarding your choices.

We thank the reviewer for highlighting these concerns. We agree that the level of detail detracts from the flow, and we have thus reduced the methodological details from the Results section and moved aspects to the Methods. Specifically, we have integrated the section on ICA denoising and single trial regressors with the Methods and have reduced their explanation in the Results section to one paragraph within the data quality section. However, since this is a dataset paper, we thought it would be important to keep in details such as head motion parameters.

The 66-dimensional embedding of the behavioral Results section comes in between fmri/meg data reliability and fmri/meg decoding. I think you were going for reliability/data quality measures for all 3 datasets, then results of richer analyses. But I'm wondering if maybe you can think of decoding quality as part of data quality, and keep those fMRI/EEG sections together more? Or, maybe signpost this Results section organization more?

The reviewer is correct that we were going for reliability and data quality for all 3 datasets, followed by results of example analyses for all three datasets. We had, indeed, tried a similar structure to the one proposed by the reviewer but then ended up deciding that the current structure is better suited, since readers may be wondering at what point we would be talking about data quality for behavior, or about the behavioral data set at all. However, we completely agree that it makes sense to signpost the Results section organization more, which we now do with a separate paragraph in the revised manuscript and with adapted subheadings.

Reviewer #3 (Recommendations for the authors):I do not have major concerns to raise for this manuscript, however, I strongly believe that the work presented would be even more valuable if the code attached to the manuscript could be improved for clarity and follow "good enough" standards for research software (https://journals.plos.org/ploscompbiol/article?id=10.1371/journal.pcbi.1005510). I know that software is not yet a valuable academic output compared to manuscripts, but I believe that the impact that will have on the community – along with the shared data – will be beyond the results and descriptions of this manuscript. In the list of comments below, I outline a minimal set of improvements to make the software more reusable. I am aware that researchers are not expected to be software engineers and I do understand if some of the requests might be too difficult, so please do as much as you can according to your skills.

We thank the reviewer for checking this critical aspect of our datasets. Indeed, the value depends strongly on the usability of the datasets, and the code is an important part that can facilitate this use a lot. In response to their comment, a major change we made was to move all code for fMRI and MEG data to GitHub, which will also make it easier to keep up with versions and to work on issues raised by the community. For simplicity, we decided to keep the code related to the behavioral dataset on OSF. We made sure to cross-reference between GitHub, Figshare, and OSF.

More generally, our aim is to address a large number of comments raised by the reviewer in this revision. Beyond the revision, we see this task of usability as an improvement that will (and should!) continue beyond this dataset paper. Even when we choose not to address a specific comment, we hope that the reviewer can see we truly care about these issues and that we intend on making THINGS-data a valuable resource for the research community.

In this context, we would like to mention that in the process of improving the MEG code, we found a minor error in our preprocessing script, and when checking the behavior code, we found another minor error in the embedding matrix. Specifically, for MEG the sensor exclusion had not been done as described in the methods section, and very few sensors had actually been excluded. We therefore decided to re-run all analyses and recreate all figures, this time without excluding any sensors. No results or conclusions changed as a consequence of skipping this preprocessing step. We have also adapted the methods section accordingly to reflect this description.

For behavior, when running the study on Mturk, there was a small mistake in the ordering of 21 out of the 1854 objects, which has to do with the different alphabetical sorting of different operating systems and programming environments. We were aware of this mistake and initially corrected it after creating the embedding. For the current dataset, since it is made publicly available, we decided to already correct this mistake at the level of the dataset, but then forgot that we did not need to correct this again after the embedding had been created. This led to a double correction, reintroducing the original mistake. Since this double correction was applied to all analyses, we had not noticed this mistake earlier during debugging, and as a consequence, it only affects a subset of the results. Specifically, the within-category analyses in Figure 3b were slightly improved for one of the conditions. We adapted the figure and the related statistics in the updated manuscript. The analyses involving RSA using behavioral similarity and fMRI/MEG led to almost identical results with no visible changes in the figures, so we kept the original figures.

In addition, we went through the 66 dimensions a second time and have slightly updated the labels, which is reflected in the updated Figure 3—figure supplement 1 (formerly Supplemental Figure 7).

List of improvements for the attached software1. LicensesLicenses are important because they let future users know if they can re-use your code. There are two zip files for the code in your manuscript. One contains licenses of the used packages and license for your code, the other did not mention any license. Please add at least a license for your code. If you are unsure, I recommend Apache 2.0 https://www.apache.org/licenses/LICENSE-2.0

We have added a CC0 license – the least restrictive kind – to all software and data. CC0 means no attribution is required when reusing materials.

2. READMECurrently, there is a readme in PDF in one zip file describing the MEG scripts. There is no readme for the fmri part. There is no readme for the other zip file. Please include a REAME file in each of the two repositories (a text file is better than PDF, in general use PDF only for figures. Any text can be a text file). The readme file should list all files included in the repository and a short explanation of what they do

We have included / improved the README files in the GitHub repository. The README instructs about data download and contains MEG and fMRI-specific code. The fMRI section additionally contains notes and examples on how to use the data and run analyses.

3. Folder structureright now there is a mixture of scripts/results/other files to prepare containers/other files from other packages all in the same folder. Please consider separating the files into meaningful subfolders. There are no rules beyond the "good enough practices" paper linked above. At least separating results/derivative files from the code would be helpful.

We agree with the reviewer. So far, we have addressed this point for the MEG analyses. All code is now command-line callable requiring the user to input the location of the BIDS-data directory. All analysis results and figures are then saved within the derivatives subfolder. For fMRI, this is currently a little challenging to implement. We are in the process of adjusting code to point to files in meaningful subfolders.

4. Dependencies:there are no dependencies listed. You need to specify which version of Matlab was used, you need to add the environment.yaml file for the conda environment that you activate in one script (if you are unsure please rune "conda env export > env.yml" after activating that environment). You need to confirm that all python scripts were run with that environment (conda activate is only present in the reconall script). You need to include the version of docker you used and the version of neurodocker. Since docker and/or neurodocker might change in the future, it would be recommended to also add the docker image you obtained to the repository or push it to dockerhub for people to reuse your docker images without needing to build them.

We have added dependencies and now specify versions more clearly. However, we decided not to push a docker image to dockerhub. We agree this is potentially useful but we believe that there will likely be many more changes to the code base in the future, which may make this step unnecessary for the purpose described by the reviewer, since the code will be kept up to date.

5. Testing and how to runYou need to document how a user can run and test the scripts themselves. For example, I have noticed that the script assumes that the BIDS (?) data should be located at a few parent subfolders. You could reconsider simplifying the work of the users, create a subfolder "BIDS" and tell the user to put there the data from neurovault and make your scripts point to the BIDS subfolder.

We agree with the reviewer and have adjusted the code accordingly. We made Python scripts command line executable and gave users the option to specify the path to the BIDS-formatted data. We will continuously develop the code and documentation and look forward to more helpful feedback from users.

6. Accepting contributions and improvements from the community + version controlThis is totally optional: right now it is difficult for a member of the community to recommend changes to your code because your code is not stored in a version control system (github, gitlab, etc). Please consider storing the code on github (or gitlab) and engage with your community of users by encouraging them to improve your code, and add future analysis to the same repository. In the long run, your repository (along with the dataset) could be a very valuable resource for the community, especially if you start getting contributions and code from future reuses of the data. Furthermore, by using a git repository you will also have the added benefits of version control of your software. Unlike manuscripts, the software is dynamic and so are the data, you can keep on improving some of your analysis or functions or just make the code more reusable by other scientists and the changes made will be documented automatically by the git system.

We completely agree with the reviewer, and – as mentioned above – we have moved the code related to the fMRI and MEG datasets to GitHub. For simplicity, we kept the code for the behavioral data on OSF. Please note that the behavioral data is much simpler to analyze, and in its preprocessed form may not require much separate code. Thus, the existing code in relation to behavior is mostly useful for reproducing the basic analyses shown in the manuscript.